# Nanomechanical action opens endo-lysosomal compartments

Yu Zhao[1], Zhongfeng Ye[1], Donghui Song[1], Douglas Wich[1], Shuliang Gao[1], Jennifer Khirallah[1] & Qiaobing Xu [1] ✉

Endo-lysosomal escape is a highly inefficient process, which is a bottleneck for intracellular delivery of biologics, including proteins and nucleic acids. Herein, we demonstrate the design of a lipid-based nanoscale molecular machine, which achieves efficient cytosolic transport of biologics by destabilizing endo-lysosomal compartments through nanomechanical action upon light irradiation. We fabricate lipid-based nanoscale molecular machines, which are designed to perform mechanical movement by consuming photons, by co-assembling azobenzene lipidoids with helper lipids. We show that lipid-based nanoscale molecular machines adhere onto the endo-lysosomal membrane after entering cells. We demonstrate that continuous rotation-inversion movement of Azo lipidoids triggered by ultraviolet/visible irradiation results in the destabilization of the membranes, thereby transporting cargoes, such as mRNAs and Cre proteins, to the cytoplasm. We find that the efficiency of cytosolic transport is improved about 2.1-fold, compared to conventional intracellular delivery systems. Finally, we show that lipid-based nanoscale molecular machines are competent for cytosolic transport of tumour antigens into dendritic cells, which induce robust antitumour activity in a melanoma mouse model.

After entering cells, most biologics are entrapped in endo-lysosomal compartments, resulting in the inability to enter the cytoplasm to function. Cationic lipid nanoparticles (LNPs) allow the cytosolic transport of these biologics by destabilizing the endo-lysosomal membranes but have low efficiency (<2%) due to the limited compartment disruption capability[1,2]. The efficiency can be improved by increasing the (positive) surface charge density of cationic LNPs[2]. However, this change usually induces the destabilization of plasma membranes during internalization, resulting in severe cytotoxicity. Inspired by biological systems, we recently reported the use of the nanomechanical action from azobenzene (Azo) derivatives to mimic the "off–on" switching of channel proteins on the plasma membrane, which allowed for controlled transmembrane transport of small molecules[3]. Therefore, abandoning the traditional viewpoints of drug delivery and finding answers from nature may be a better approach to overcome the above-mentioned issues[4,5]. Molecular machines are assemblies of a certain number of molecular components (usually proteins), which perform specific mechanical movements (outputs) through changing their conformation in response to appropriate external stimuli (inputs)[6,7]. These molecular machines enable complex and delicate processes by executing rotational and translational movements at the cellular level, which are essential for regulating intracellular, transmembrane, and intercellular transport of substances[8–10]. Kinesin, dynein, and their relatives are amazing examples of molecular machines. They convert chemical energy from adenosine-5′-triphosphate (ATP) hydrolysis into mechanical work along cellular microtubules for macromolecule transport[11]. Unfortunately, these natural molecular machines can hardly handle the transport of exogenous biologics within cells, not to mention the transmembrane transport from endo-lysosomal compartments to the cytoplasm.

Herein, we design and fabricate an artificial nanoscale molecular machine (nanomachine) by co-assembling photoisomerable amphiphilic Azo-based lipidoids and helper lipids, which achieves enhanced cytosolic transport of exogenous biologics by disrupting endo-lysosomal compartments through the nanomechanical action of

---

[1]Department of Biomedical Engineering, Tufts University, Medford, MA 02155, USA. ✉e-mail: qiaobing.xu@tufts.edu

azobenzene upon light irradiation. Unlike natural molecular machines, this lipid-based nanomachine (LNM) is designed to convert light energy into mechanical movements[12]. This is because photons are one of the most convenient energy inputs to drive LNMs at a high spatiotemporal resolution. Azo-based lipidoids were synthesized to perform light-driven mechanical movement, in which its operation was based on an efficient and clean photoreaction, namely, photoisomerization[13–15]. Specifically, Azo-based lipidoids in LNMs undergo the reversible isomerization between compact *cis*- and extended *trans*-isomers upon irradiation with ultraviolet (UV, 365 nm) and visible light (Vis, >400 nm, Fig. 1a). The isomerization of Azo-based lipidoids caused by simultaneous UV and Vis (UV/Vis) irradiation enables them to produce continuous rotation−inversion and stretch−shrink movements. Thus, after entering cells via endocytosis, LNMs interact with the endo-lysosomal membrane, and Azo-based lipidoids act as rotors upon irradiation with UV/Vis light, thereby enhancing destabilization and disruption of the membrane through nanomechanical action (Fig. 1b). In this work, a lipid formulation is chosen due to its capability in loading various cargoes. This architecture allows for efficient transport of exogenous biologics from endo-lysosomal compartments to the cytoplasm driven by the consumption of photons (inputs). Our results show that LNMs can enhance endo-lysosomal transport (escape) and facilitate efficient delivery of various biologics into the cytoplasm. We find that both nucleic acid (e.g., mRNAs) and protein cargoes (e.g., Cre proteins) can be transported to the cytoplasm using this LNM-based biomimetic approach, which provides a solution to improve the intracellular delivery of biologics for therapeutic applications. To demonstrate the generality of LNMs and broaden their potential applications, we further investigate their capability to carry tumour antigen from endo-lysosomal compartments to the cytoplasm and present the antigen on major histocompatibility complex class I (MHC-I) in dendritic cells (DCs). This is essential for achieving efficient antigen cross-presentation to enhance DC vaccine-based immunotherapy. We find that LNMs can facilitate efficient tumour antigen cross-presentation and DC maturation when coupled with UV/Vis light irradiation. The results from animal experiments show that administration of irradiated DCs pre-treated with LNM/tumour antigen complexes triggers enhanced cross-priming of cytotoxic T cells, thereby achieving robust antitumour immunity in a melanoma mouse model.

## Results

### Characterization of Azo-based lipidoids and lipid-based nanomachines

First, two different photoisomerable amphiphilic Azo-based lipidoids, SAzo and TAzo lipidoids, were synthesized (Supplementary Figs. 1 and 3−5). As demonstrated in Fig. 1a, both lipidoids consist of Azo quaternary ammonium salts as the hydrophilic headgroup and flexible alkyl chains as the hydrophobic tail, where the SAzo lipidoid has a single tail and the TAzo lipidoid has two tails. We then investigated the photoisomerization kinetics of SAzo lipidoids (30 μM) and TAzo lipidoids (15 μM) in dimethyl sulfoxide (DMSO) solutions using both ultraviolet and visible (UV−Vis) absorption spectrophotometry. As shown in Fig. 2b and c, when SAzo and TAzo lipidoids were exposed to UV light (365, 5.40 mW/cm²), the absorbance at 360 nm decreased gradually (corresponding to the $\pi-\pi^*$ transition in the *trans*-form) with the increase of UV irradiation time, while the absorbance at 440 nm increased (corresponding to the $n-\pi^*$ transition in the *cis*-form)[16]. Further experimentation showed that these spectral changes levelled off after 6 s of UV exposure due to the establishment of the photo-stationary state (Supplementary Fig. 6a and b). Subsequently, an additional 6 s of Vis light (>400 nm) irradiation reversed these spectral changes, suggesting that the Azo units underwent rapid isomerization, even after modification with quaternary ammonium groups and alkyl chains. During the process of *cis−trans* isomerization, irradiation intensities were crucial parameters that directly determined the isomerization rates. In order to create and maintain a continuous rotation−inversion movement with these Azo units, similar *trans*-to-*cis* and *cis*-to-*trans* isomerization rates are necessary. This is because SAzo and TAzo lipidoids tend toward a photostationary state if the isomerization rates are not in equilibrium. According to the results, there was an optimal intensity of Vis light (20 mW/cm², 3.7 times that of UV intensity) for achieving similar isomerization rates (Supplementary Fig. 6c and d). This difference between UV and Vis light intensities could be attributed to the fact that Vis-triggered *cis*-to-*trans* isomerization is more difficult than UV-triggered *trans*-to-*cis* isomerization[17].

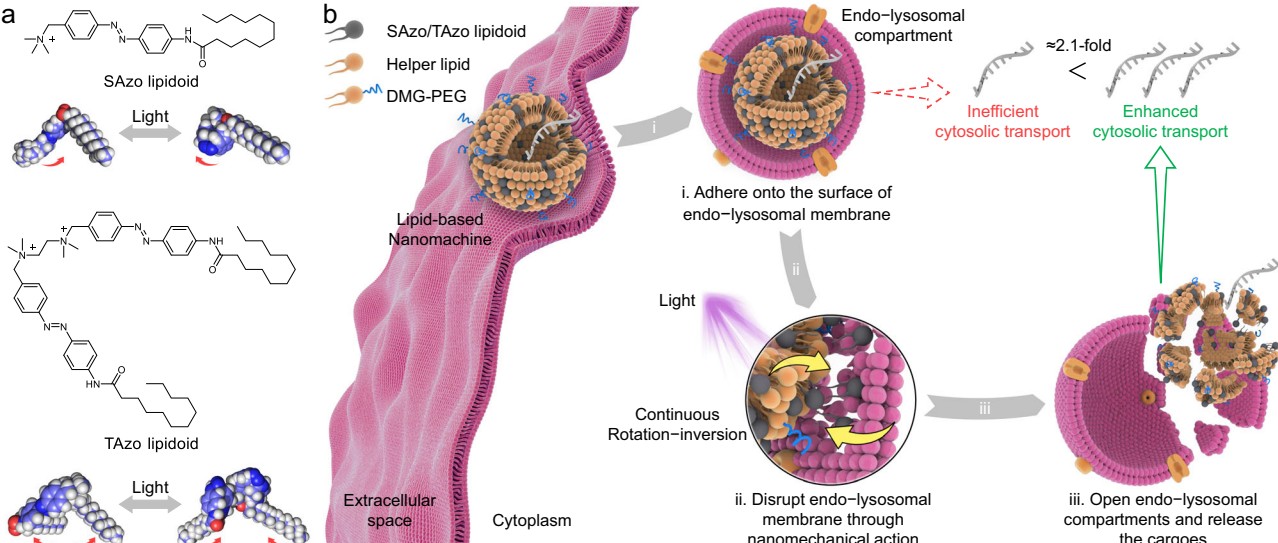

**Fig. 1 | Lipid-based nanomachine (LNM) opens endo-lysosomal compartments through nanomechanical action. a** Chemical structures of the two different Azo-based lipidoids, including SAzo lipidoid with a single tail and TAzo lipidoid with two tails. **b** Schematic illustration of the LNM structure and its potential mechanism of overcoming intracellular barriers through light-triggered nanomechanical action. (i) The LNM adheres onto the inner surface of the endo-lysosomal membrane through electrostatic interaction after entering cells via endocytosis. (ii) Reversible isomerization of the Azo unit induced by simultaneous UV and Vis light irradiation causes Azo-based lipidoids to undergo continuous rotation−inversion and stretch−shrink movements. Thus, LNMs containing Azo-based lipidoids destabilize the endo-lysosomal membrane and induce membrane disruption. (iii) LNM opens the endo-lysosomal compartments and transports the cargoes to the cytoplasm.

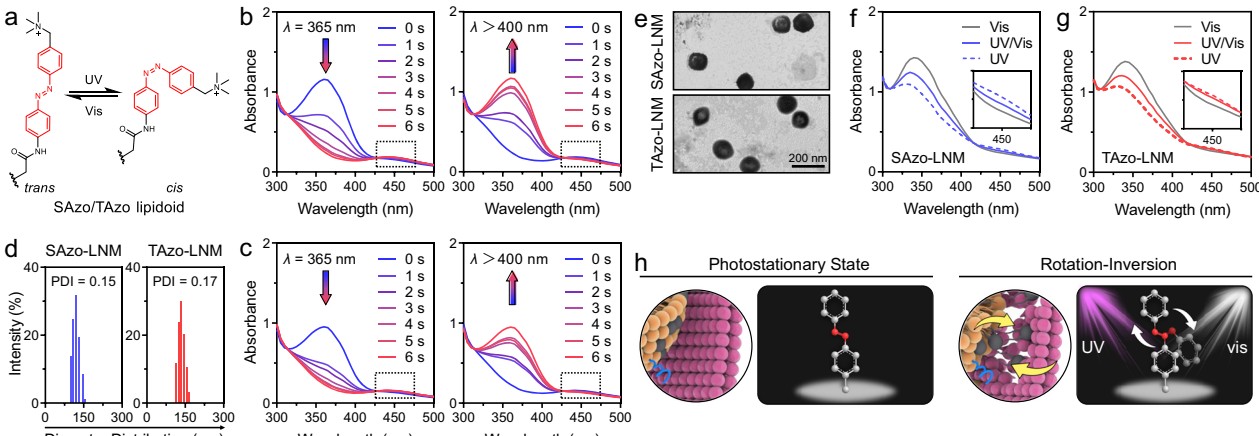

**Fig. 2 | Photoisomerization of SAzo and TAzo lipidoids. a** Schematic illustration of the *trans*-to-*cis* and *cis*-to-*trans* photoisomerization of an Azo unit when irradiated with different lights. **b** UV−Vis absorption spectrum of SAzo lipidoids before and after irradiation with UV (365 nm for 6 s) and Vis light (>400 nm for 6 s), respectively. **c** UV−Vis absorption spectrum of TAzo lipidoids before and after irradiation with UV (365 nm for 6 s) and Vis light (>400 nm for 6 s), respectively. The power densities for 365 and >400 nm are 5.40 and 20 mW/cm², respectively. These studies were performed in DMSO solution. **d** Diameter distributions of SAzo-LNM and TAzo-LNM in PBS (10 mM, pH 7.4) measured by DLS, together with the polydispersity index (PDI). SAzo-LNM formulation: the molar ratio of SAzo lipidoid:DOPE:Chol:DMG-PEG was 30:30:36.5:3.5. TAzo-LNM formulation: the molar ratio of TAzo lipidoid:DOPE:Chol:DMG-PEG was 15:30:36.5:3.5 (30:30:36.5:3.5, Azo units as standard). **e** Representative TEM images of SAzo-LNM and TAzo-LNM. Scale bar, 200 nm. **f** and **g** UV−Vis absorption spectrum of SAzo-LNM and TAzo-LNM after different treatments. The samples are irradiated with Vis, UV and UV/Vis light for 180 s in PBS at 37 °C, respectively. **h** Schematic illustration of continuous rotation−inversion of SAzo or TAzo lipidoid caused by simultaneous UV (purple) and Vis (white) light irradiation.

Next, two different LNMs were prepared by co-assembling SAzo or TAzo lipidoids with 1,2-dioleoyl-sn-glycero-3-phosphoethanolamine (DOPE), cholesterol (Chol), and 1,2-dimyristoyl-rac-glycero-3-methoxypolyethylene glycol-2000 (DMG-PEG) (denoted as SAzo-LNM and TAzo-LNM, respectively)[18–20]. Dynamic light scattering (DLS) measurements showed that SAzo-LNMs and TAzo-LNMs had similar diameters of ~130 nm in phosphate-buffered saline (PBS, 10 mM, pH 7.4) (Fig. 2d). Both LNMs exhibited spherical morphologies as observed by transmission electron microscopy (TEM) (Fig. 2e). Next, their photoisomerization kinetics were measured in PBS at 37 °C. According to the results, these two LNMs easily reached the *cis*- or *trans*-rich photostationary states after being irradiated with UV or Vis light for about 30 s, respectively (Supplementary Fig. 6e and f). Compared to the time required (6 s) in DMSO solutions, the longer time might be attributed to the relatively crowded environment in an assembly formulation.

To investigate the continuous rotation−inversion movement, LNMs were exposed to UV, Vis, and UV/Vis light for 180 s, respectively, which was six times longer than the time required to reach the photostationary states. As shown in Fig. 2f and g, long-time UV irradiation enabled SAzo and TAzo lipidoids to reach a *cis*-rich state (low absorbance at 340 nm, coloured dotted lines), while Vis irradiation enabled them to reach a *trans*-rich state (high absorbance at 340 nm, grey solid lines). When this sample was exposed to UV/Vis light for 180 s, moderate-intensity absorptions (colored solid lines) were observed. These results suggested that co-irradiation with UV and Vis light led to simultaneous *cis*- and *trans*-isomerization of Azo-based lipidoids to reach and maintain a metastable equilibrium state containing nearly equal amounts of *cis*- and *trans*-forms, rather than a photostationary state. In this metastable equilibrium state, a certain amount of *trans*-isomers should be converted into the *cis*-forms because of UV irradiation, while a similar amount of *cis*-isomers should be converted into *trans*-forms from Vis irradiation. During this process, Azo-based lipidoids in LNMs should undergo continuous rotation−inversion and stretch−shrink movements in the presence of UV/Vis light, as demonstrated in Fig. 2h.

We further confirmed the dynamic isomerization of Azo-based lipidoids by quantitative analysis of their ¹H NMR spectra. As demonstrated in Supplementary Fig. 6g, we irradiated the Azo-based lipidoids with different irradiation combinations for 180 s. Then, we removed the light and acquired the ¹H NMR spectrum immediately. Considering that both *cis*-isomers and *trans*-isomers reached the thermodynamic equilibrium state via thermal relaxation at a much slower rate (>48 h, Supplementary Fig. 6h and i), Azo-based lipidoids could maintain a specific configuration for a period after removing the light. Thus, the acquired ¹H NMR spectrum reflects the instantaneous percentage of *cis*- and *trans*-isomers during irradiation. Supplementary Fig. 6g showed that the combinations of different light intensities enabled the isomerization of Azo-based lipidoids to reach and maintain different equilibrium states. Specifically, increasing the UV light intensity resulted in a shift of the isomerization equilibrium towards the *cis*-rich state, whereas increasing the Vis light intensity shifted the equilibrium towards the *trans*-rich state. These results demonstrated that Azo-based lipidoids underwent dynamic isomerization during light exposure. As a result, Azo-based lipidoids show the potential to function as rotors to destabilize and disrupt the endo-lysosomal membrane through nanomechanical action, thereby enabling efficient trans-membrane transport of biologics within cells (Fig. 1b).

## LNM facilitates cytosolic transport of mRNA both in vitro and in vivo

Various clinical applications can be achieved through the delivery of mRNAs expressing antigens of cancers or infectious diseases, as well as disease-related therapeutic proteins[21,22]. The recent success of two mRNA vaccines produced by Pfizer/BioNTech and Moderna for preventing coronavirus 2019 (COVID-19) further highlights the enormous potential of mRNA-based therapies to revolutionize life science and medical research[23–27]. Thus, as a proof-of-concept, we chose green fluorescent protein (GFP) mRNA as a model cargo and studied the capability of LNMs to enhance transfection efficiency. A formulation without Azo-based lipidoids was prepared as a negative control, where 1,2-dioleoyloxy-3-trimethylammonium propane chloride (DOTAP) that also has a quaternary ammonium salt group was employed as an alternative to the Azo-based lipidoid (denoted as DOTAP-LNP). The cytotoxicity of LNMs was first examined in HeLa cells. As shown in Fig. 3a, negligible cytotoxicity of LNMs was observed at concentrations below 500 µg mL⁻¹ after a 24-h incubation period, suggesting relatively

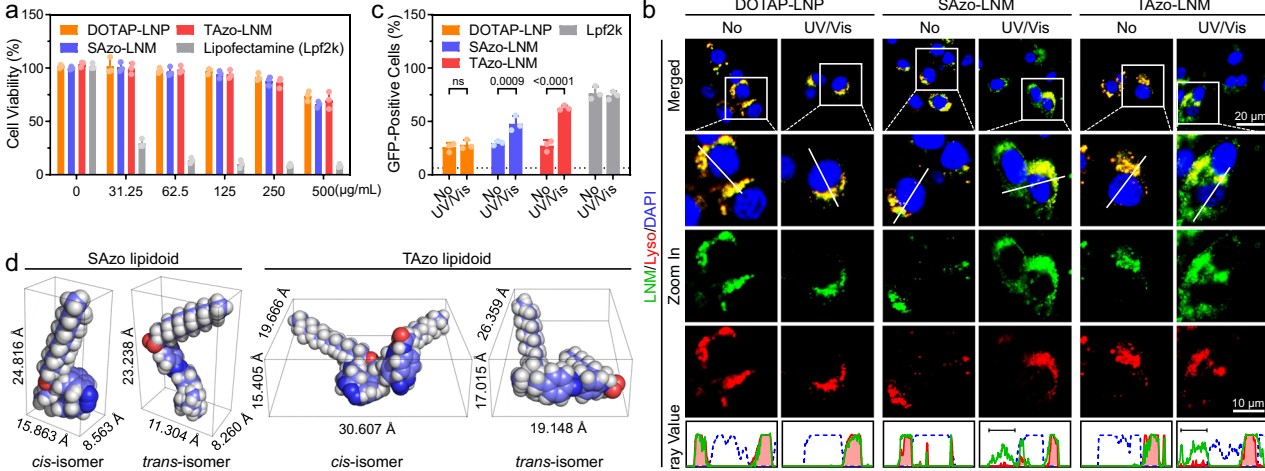

**Fig. 3 | LNM transports mRNA from endo-lysosomal compartments to cytoplasm. a** Cell viability of HeLa cells after 24-h incubation with various concentrations of DOTAP-LNP, SAzo-LNM, TAzo-LNM, and Lpf2k, respectively. **b** Fluorescence images of the cytosolic transport capabilities of DOTAP-LNP, SAzo-LNM, and TAzo-LNM in HeLa cells, respectively. Fluorescent-labelled formulations are prepared (green) and incubated with cells. After 1 h of incubation (allowing these formulations to enter cells), the cells were irradiated with UV/Vis light for 10 min. After another 30 min incubation, LysoTracker Red was added to stain the endo-lysosomal compartments. Scale bar, 10 μm. **c** GFP mRNA transfection efficacy of DOTAP-LNP, SAzo-LNM, TAzo-LNM, Lpf2k, and naked GFP mRNA (1 μg mL⁻¹, dotted line) tested on HeLa cells by quantifying GFP-positive cells. **d** Molecular dimensions of SAzo lipidoid and TAzo lipidoid. Data are presented as mean ± standard deviation (s.d.) from *n* independent experiments (*n* = 3). Statistical significance was analysed by two-way ANOVA with Sidak's multiple comparisons test. *P* values are indicated.

low cytotoxicity of these synthetic formulations. In contrast, the commercial transfection reagent Lipofectamine 2000 (Lpf2k) showed significant cytotoxicity under the same incubation conditions.

Next, we studied the capability of LNMs to reach the cytoplasm from endo-lysosomal compartments. Fluorescent-labelled SAzo-LNMs, TAzo-LNMs, and DOTAP-LNPs were prepared by using L-α-phosphatidylethanolamine-N-(4-nitro-benzo-2-oxa-1,3-diazole) (PE-NBD, a fluorescent lipid). These LNMs were then incubated with HeLa cells for 1 h, which was sufficient for them to be taken up by cells (Supplementary Fig. 8a). Then, the cells received UV/Vis light irradiation for 10 min. After another 30 min incubation, the cells were stained with LysoTracker Red and observed by confocal laser scanning microscopy (CLSM). As demonstrated in Fig. 3b, SAzo-LNM and TAzo-LNM (green) could efficiently reach the cytoplasm from endo-lysosomal compartments (red) upon exposure to UV/Vis light. In the absence of UV/Vis light, these LNMs remained trapped in the endo-lysosomal compartments[28,29]. Moreover, live-cell imaging experiments showed that the transmembrane efficiencies of these fluorescent-labelled LNMs were irradiation time-dependent (Supplementary Fig. 7a and b)[30]. With an increase in irradiation time, the signal of LNMs in the endo-lysosomal compartments decreased, while the signal of LNMs in the cytoplasm increased. In contrast, almost all DOTAP-LNPs were entrapped in endo-lysosomal compartments regardless of the use of UV/Vis irradiation.

To further confirm the above results, co-localization analysis based on CLSM images was performed. Pearson's co-localization coefficient (PCC) between the fluorescence signals of different PE-NBD-labelled formulations and LysoTracker Red were calculated, respectively. PCC would be close to 1 if these two signals are highly co-localized. According to the results, the PCC values in the SAzo-LNM and TAzo-LNM groups are lower in the presence of UV/Vis irradiation (PCC$_{SAzo-LNM}$ = 0.80; PCC$_{TAzo-LNM}$ = 0.71), compared with these samples without irradiation (PCC$_{SAzo-LNM}$ = 0.90; PCC$_{TAzo-LNM}$ = 0.87, Supplementary Fig. 8b and c). In contrast, the PCC value in the DOTAP-LNP group is close to 1 (PCC$_{DOTAP-LNP}$ = 0.93) even when exposed to UV/Vis light. These results confirmed the light-enhanced endo-lysosomal escape effects and the crucial role of Azo-based lipidoids in light-induced membrane disruption. Interestingly, we found that the PCC value in the TAzo-LNM group is lower than that of the SAzo-LNM

group, suggesting that TAzo-lipidiods may have a higher endosomal escape efficiency. This speculation is further corroborated in the following transfection efficiency and mechanism studies.

We then investigated the capability of LNMs to facilitate GFP mRNA transfection by disrupting the endo-lysosomal compartments with light irradiation. The particle size and zeta potentials of GFP mRNA-loaded LNMs were first examined (Supplementary Fig. 9c). Different formulations loaded with GFP mRNA at five different nitrogen/phosphorus (N/P) molar ratios (from 0.5 to 10) were prepared using the ethanol dilution method[31]. As shown in Supplementary Fig. 9a and b, a slight increase in particle size was observed when loaded with GFP mRNA. In terms of zeta potentials, SAzo-LNM and TAzo-LNM followed the same profile in function of the N/P ratio. At low N/P ratios, such as 0.5 and 1.25, the formulations were negatively charged, while the charge proportionally increased with the increase of N/P ratio. Based on these results and previous studies, an N/P molar ratio of 2.5 was selected for further experiments[32]. Moreover, we found that mRNA loading did not affect the light-controlled reversible photoisomerization of Azo-based lipidoids (Supplementary Fig. 9d). Next, different mRNA-loaded formulations were incubated with cells for 4 hours, then irradiated with light for 10 min. Lpf2k and naked GFP mRNA were used as positive and negative controls, respectively. As shown in Fig. 3c and Supplementary 10a, light irradiation resulted in a 1.6-fold and 2.3-fold increase in transfection efficiency in the SAzo-LNM and TAzo-LNM groups, respectively. The transfection efficiency of the TAzo-LNM group is comparable to that of Lpf2k. However, relatively low transfection efficiencies and negligible differences were observed in DOTAP-LNP group with or without irradiation. These results suggested that light irradiation improved the efficiency of LNMs containing Azo-based lipidoids for transporting GFP mRNA into the cytoplasm for GFP expression. Moreover, we found that the mRNA-loaded LNMs showed negligible cytotoxicity against HeLa cells (>90%, mRNA, 1 μg mL⁻¹; formulation, 50 μg mL⁻¹), while the mRNA/Lpf2k complex showed significant cytotoxicity (32.0%, Supplementary Fig. 10b). These results showed good tolerability of these LNMs.

We then conducted a study on the molecular dimensions of SAzo and TAzo lipidoids in their *cis-* and *trans-*conformations (Fig. 3d). The results showed that TAzo lipidoids undergo greater changes in their

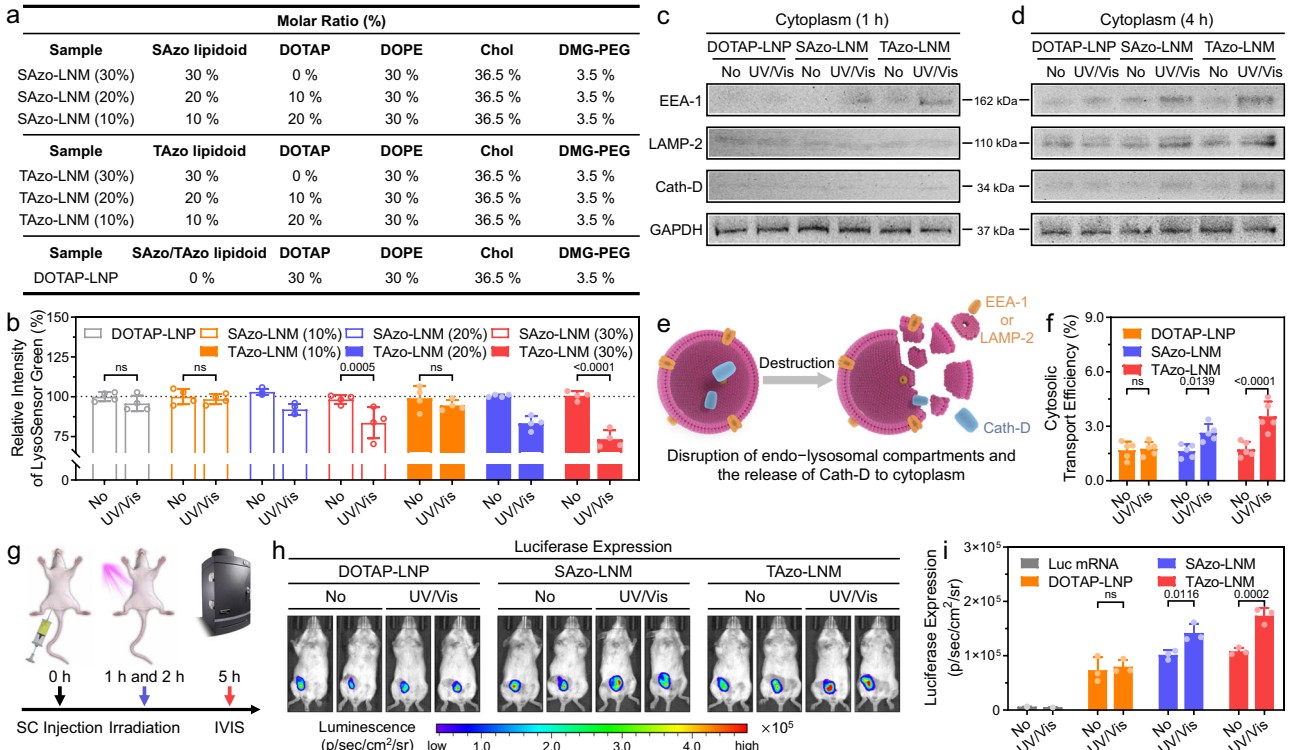

**Fig. 4 | LNM destabilizes the endo-lysosomal membrane and induces membrane disruption. a** Formulation details of LNMs with various percentages of SAzo and TAzo lipidoids (molar ratio, Azo units as standard). **b** Fluorescence intensity of LysoSensor Green in HeLa cells after incubating with different LNMs and treating with or without UV/Vis light irradiation. **c** and **d** Detection of EEA-1, LAMP-2, and Cath-D in the cytoplasm of HeLa cells after different treatments using western blotting analysis. Three times experiments are repeated with similar results. **e** Schematic illustration of disruption of endo-lysosomal compartments and the release of EEA-1, LAMP-2, and Cath-D to the cytoplasm. **f** Analysis of cytosolic transport efficiency of different formulations by SLEEQ assay. **g** Schematic illustration of the experimental design. The mice were irradiated with UV/Vis light for 15 min at 1 and 2 h after subcutaneous (SC) injection of mRNA-loaded LNMs. The power densities for 365 and >400 nm are 30 and 120 mW/cm², respectively. **h** Representative whole-body bioluminescence images of mice after SC injection of different formulations measured by the IVIS imaging system (0.25 mg kg⁻¹ Luc mRNA, 5 µg per mouse). Images were taken at 5 h post-injection, and two mice in each group were shown. **i** Relative luciferase expression in each group. Data are presented as mean ± standard deviation (s.d.) from *n* independent experiments (**b**, *n* = 4; **f**, *n* = 5; **i**, *n* = 3). Statistical significance was analysed by two-way ANOVA with Sidak's multiple comparisons test. *P* values are indicated.

molecular height, length, and width compared to SAzo lipidoids during isomerization. Such dramatic changes in molecular dimensions may lead to increased disruption of endo-lysosomal membranes upon light-induced nanomechanical action of LNMs.

To further confirm this theory, a series of SAzo-LNMs and TAzo-LNMs with reduced percentages of Azo-based lipidoids (20% and 10%) were prepared, respectively (Fig. 4a). DOTAP was employed in these formulations as an alternative to Azo-based lipidoids to ensure that each formulation contained equivalent quaternary ammonium salt groups, and had similar physical properties. In this study, cells were incubated with different formulations, exposed to UV/Vis light, and stained by LysoSensor Green[33]. It is shown that destabilization or disruption of the endo-lysosomal compartments typically suppresses the LysoSensor signal[34,35]. This is because LysoSensor probes rely on the decreased pH values to sense the endo-lysosomal compartments, and the protons that create a low pH environment are released into the cytoplasm upon the compartment rupture[36,37]. Moreover, LysoSensor probes exhibit a pH-dependent increase in fluorescence intensity and are almost non-fluorescent when outside acidic compartments[38]. Based on this, we studied the integrity of endo-lysosomal compartments by measuring the fluorescence intensity of LysoSensor Green. As shown in Fig. 4b, the fluorescence intensity in the TAzo-LNM (30%) group decreased to 73.4% after exposure to UV/Vis light compared to without light. And the fluorescence intensity decreased to 83.6% in the SAzo-LNM (30%) group under the same conditions. These results showed that a more distinct destabilization of endo-lysosomal

compartments occurred in the TAzo-LNM (30%) group compared to that of the SAzo-LNM (30%) group, which is consistent with the mRNA transfection efficiency results (Fig. 3c). Moreover, when Azo-based lipidoid percentages in LNMs decreased from 30% to 10%, the compartment destabilization was gradually reduced, further confirming the critical role of Azo-based lipidoids in LNM-mediated transmembrane process.

To further investigate the effects of light irradiation on endosomal and lysosomal compartment disruption, we utilized western blot (WB) immunoassay to evaluate whether specific protein markers were released from the endosomal (early endosomal antigen-1 (EEA-1)) and lysosomal (lysosome-associated membrane protein 2 (LAMP-2)) compartments to the cytoplasm[39,40]. The cells were incubated with different LNMs for 1 or 4 h, respectively, and then exposed to UV/Vis light. We observed that the release of both EEA-1 (Fig. 4c and Supplementary Fig. 11a) and LAMP-2 (Fig. 4d and Supplementary Fig. 11b) were enhanced in the LNM groups irradiated with light. In contrast, there was no significant difference in protein release between DOTAP-LNM groups that did or did not receive light irradiation. Our results also showed that incubation of cells with SAzo-LNMs and TAzo-LNMs for 1 h coupled with UV/Vis irradiation only resulted in the release of EEA-1 into the cytoplasm, and incubation of cells for 4 h significantly enhanced the release of LAMP-2. It is known that nanoparticles enter endosomal compartments upon being taken up by cells in the first 1–2 h, and are then transferred into the lysosomal compartments after 4 h[1]. This explains why we observed the enhanced EEA-1 release after

1 h of incubation, followed by enhanced LAMP-2 release after 4 h of incubation. The release of Cathepsin D (Cath-D), a lysosomal protease, was also evaluated to confirm the enhanced lysosomal disruption upon light irradiation. We did not observe Cath-D release in the first hour regardless of light irradiation (Fig. 4c), while increased Cath-D release was observed after 4 h of incubation (Fig. 4d), confirming the LNM-mediated lysosomal compartment disruption. These results suggested that the nanomechanical action of Azo-based lipioids could trigger the effective disruption of endosomal and lysosomal compartments. TEM images showed that UV/Vis irradiation loosened the nanostructure of LNMs possibly due to the stretch–shrink movements of Azo-based lipidoids (Supplementary Fig. 11c). The size changes of LNMs after UV/Vis irradiation were then evaluated using DLS. As shown in Supplementary Fig. 11d and 11e, UV/Vis irradiation did not cause great changes in the particle size of LNMs, but their PDI values increased significantly after irradiation. The increase in PDI values might be attributed to the nanostructural changes of LNMs induced by nanomechanical action, which is consistent with the TEM results. Agarose gel electrophoresis showed that irradiating mRNA-loaded LNMs for 10 min did not cause mRNA leakage. Specifically, we monitored the release of the mRNA from the LNMs with and without light irradiation. Free mRNA was used as a control. As demonstrated in Supplementary Fig. 10c, almost all mRNA was retained near the gel wells with or without UV/Vis irradiation in both SAzo-LNM and TAzo-LNM groups, indicating that no free mRNA was released from LNMs regardless of light irradiation. Interestingly, we observed slight band migration in UV/Vis irradiated mRNA-loaded LNMs, compared to non-irradiated samples. This slight band migration showed that the light irradiation does cause the LNM structure to loosen, but does not cause disruption. This ensured that mRNAs could be transported to the cytoplasm by LNMs, rather than being remained in endo-lysosomal compartments.

Next, to quantify the cytosolic transport efficiency of LNMs, we employed a recently reported quantification technique called the split luciferase endosomal escape quantification (SLEEQ) assay[41]. The SLEEQ assay is based on a highly sensitive bioluminescent split luciferase system that is composed of two subunits: a large BiT protein (LgBiT, 17.8 kDa) and a high-affinity complementary peptide (HiBiT, 1.3 kDa). When separated these fragments have no luminescent activity, but when brought together, they form a functional enzyme that binds to a substrate which produces bright luminescence. We first transfected HeLa cells with a LgBiT expression vector. LgBiT-expressing cells were then incubated with HiBiT-loaded LNMs for 4 h and received 10 min of UV/Vis light irradiation. The release of HiBiT into the cytoplasm generated a luminescent signal, where the signal intensity correlated with the efficiency of endo-lysosomal release. The cytosolic transport efficiency was calculated according to the formula in Supplementary Fig. 2. According to the results, the cytosolic transport efficiency of the SAzo-LNM group increased 1.6-fold from 1.6% to 2.6%, and the TAzo-LNM group transport efficiency increased 2.1-fold from 1.7% to 3.6% upon exposure to irradiation (Fig. 4f).

We further investigated the effects of light irradiation of the LNMs on the mRNA delivery efficiency in vivo. We encapsulated firefly luciferase mRNA (Luc mRNA) in different LNMs, which the expression of luciferase could be visualized in vivo using the IVIS imaging system (PerkinElmer). Luc mRNA in a DOTAP formulation was used as a control for comparison. The injection site was irradiated with UV/Vis twice (1 and 2 h after receiving the injection) (Fig. 4g). As shown in Fig. 4h and Supplementary Fig. 11f, the UV/Vis light irradiation significantly improved the bioluminescence signal in both the SAzo-LNM and TAzo-LNM groups, compared with the groups without light irradiation. In contrast, negligible differences in luminescence signal were observed in the DOTAP formulation regardless of light irradiation. Further quantitative analysis showed that UV/Vis irradiation resulted in 1.3-fold and 1.6-fold increase in transfection efficiency in the SAzo-LNM and

TAzo-LNM groups, respectively (Fig. 4i). These results suggested that mRNA could be transported to the cytoplasm with a higher efficiency using an LNM-based biomimetic approach both in vitro and in vivo.

## LNM facilitates cytosolic transport of Cre proteins

Next, we studied the potential of LNMs to transport protein cargoes. Intracellular delivery of genome-editing proteins, such as Cre recombinase (Cre) and Cas9/sgRNA ribonucleoproteins, can regulate cellular gene expression with high specificity[42–44]. Thus, we investigated the capability of LNMs to transport Cre across the endo-lysosomal membranes. (−30)GFP-Cre (obtained by fusing a negatively charged GFP variant, (−30)GFP, to Cre), HeLa cells, and HeLa-DsRed cells were used in this study[43,45]. We assembled (−30)GFP-Cre with SAzo-LNMs and TAzo-LNMs, respectively, and then incubated with HeLa cells. Naked (−30)GFP-Cre and DOTAP-LNP/(−30)GFP-Cre complexes were employed for comparison. The internalization of (−30)GFP-Cre by different formulations (protein, 1.5 μg mL⁻¹; formulation, 50 μg mL⁻¹) was first quantified by counting GFP-positive cells. Similar levels of GFP-positive cells were observed in the DOTAP-LNP, SAzo-LNM, and TAzo-LNM groups (Supplementary Fig. 12a). The intracellular trafficking of (−30)GFP-Cre was then studied by visualizing the localization of (−30)GFP-Cre using CLSM imaging. A procedure similar to mRNA transport was performed. As shown in Fig. 5a, compared to DOTAP-LNPs, treatment with SAzo-LNM/(−30)GFP-Cre and TAzo-LNM/(−30)GFP-Cre complexes led to significant accumulation of GFP signal (green) in the cytoplasm and nuclei (blue) with a lower level of co-localization in endo-lysosomal compartments (red) when exposed to UV/Vis light. Furthermore, these results were confirmed with co-localization analysis by calculating the PCC values between the fluorescence signals of (−30)GFP-Cre and LysoTracker Red (Fig. 5b). The inherent nucleus targeting capability of the Cre enabled the effective accumulation of (−30)GFP-Cre in nuclei for gene recombination.

Next, to evaluate the recombination efficiency, HeLa-DsRed cells were used. These cells are genetically integrated with a LoxP-flanked STOP cassette to prevent the transcription of red fluorescent DsRed while expressing DsRed in Cre-mediated gene recombination (Fig. 5d)[46]. LNM/(−30)GFP-Cre complexes were incubated with cells for 4 h, then received 10 min of UV/Vis irradiation, and incubated for another 20 h. Following this, we quantified the percentages of DsRed-positive cells in each group to determine the recombination efficiency. The results showed that treatment of HeLa-DsRed cells with DOTAP-LNPs, SAzo-LNMs, and TAzo-LNMs resulted in significantly enhanced DsRed expression compared to naked GFP-Cre (GFP-Cre, 5.17%; DOTAP-LNPs, 22.3%; SAzo-LNMs, 25.9%; TAzo-LNMs, 24.6%; Fig. 5c and Supplementary Fig. 12b). Importantly, UV/Vis light irradiation further improved the levels of DsRed-positive cells in the SAzo-LNM (33.9%) and TAzo-LNM (44.9%) groups, particularly the latter. These results indicated that light-driven LNMs also showed great potential to facilitate cytoplasm transport of protein-based biologics.

## LNMs facilitate cross-presentation of tumour antigens for cancer immunotherapy

Dendritic cell (DC)-based cancer vaccines have gained notable advances in recent years[47]. In this therapy, DCs are typically pre-incubated with tumour antigens or tumour cell lysates[48]. These primed autologous DCs can induce T cell-dependent antitumour responses when they migrate from the administration site to the draining lymph nodes. However, only a limited number of patients benefited from DC vaccination in clinical trials. An important reason is that the tumour antigens internalized by DCs are easily degraded within lysosomal compartments, resulting in limited cross-presentation[49]. Considering the success of LNMs to carry protein-based biologics into the cytoplasm, we next investigated whether the LNMs could transport tumour antigens into the cytoplasm to bind with MHC-I in DCs for enhanced cross-presentation.

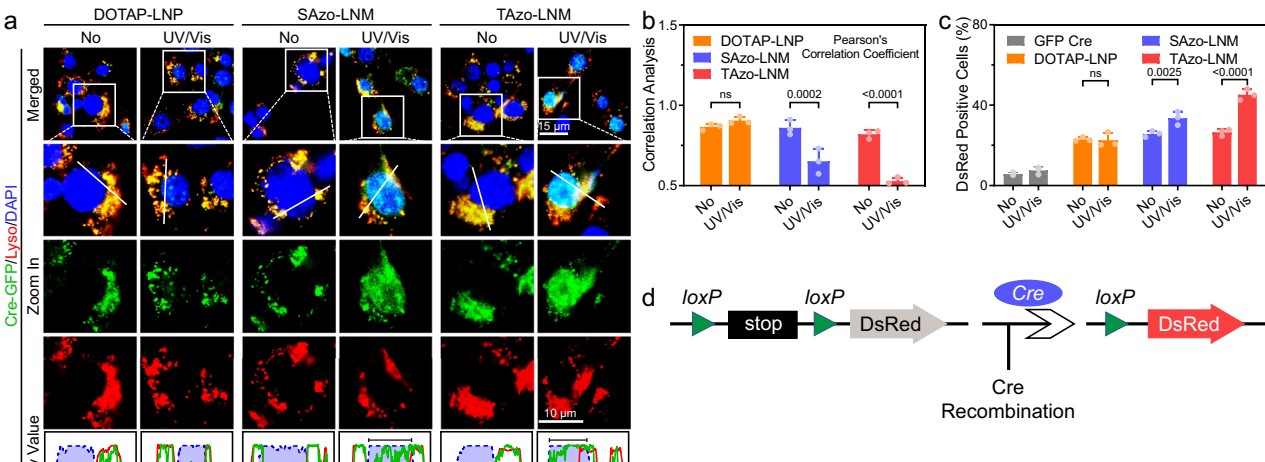

**Fig. 5 | LNM transports genome editing proteins from endo-lysosomal compartments to cytoplasm. a** Fluorescence images of HeLa cells after treatment with DOTAP-LNP/(−30)GFP-Cre, SAzo-LNM/(−30)GFP-Cre, and TAzo-LNM/(−30)GFP-Cre (1.5 μg mL⁻¹ protein), respectively. Different formulations are incubated with cells. After 30 min incubation, the cells receive UV/Vis light irradiation for 10 min. After further incubation, LysoTracker Red is added to stain the compartments. Scale bar, 10 μm. **b** Quantitative analysis of co-localization of (−30)GFP-Cre with LysoTracker Red-labelled endo-lysosomal compartments (three independent experiments). The coefficients are close to 1 if they are highly colocalized. **c** DsRed expression efficacy of DOTAP-LNP/(−30)GFP-Cre, SAzo-LNM/(−30)GFP-Cre, TAzo-LNM/(−30)GFP-Cre, and naked (−30)GFP-Cre (1.5 μg mL⁻¹ protein) tested on HeLa-DsRed cells, respectively. **d** Schematic illustration of delivery of Cre protein to delete the stop cassette and activate downstream DsRed protein. Data are presented as mean ± standard deviation (s.d.) from *n* independent experiments (*n* = 3). Statistical significance was analysed by two-way ANOVA with Sidak's multiple comparisons test. *P* values are indicated.

The impact of LNMs on antigen cross-presentation was first evaluated in murine bone marrow-derived dendritic cells (BMDCs). Ovalbumin (OVA) was used as a model antigen. LNM/OVA complexes were obtained by mixing different LNMs and OVA. These complexes were then incubated with BMDCs and received UV/Vis light irradiation (Fig. 6a). The cross-presentation of OVA in BMDCs was studied by flow cytometry assay and CLSM observation. As shown in Fig. 6b and c, BMDCs treated with SAzo-LNM/OVA and TAzo-LNM/OVA complexes exhibited 1.7-fold and 2.6-fold higher presentation of OVA fragments on MHC-I after UV/Vis light irradiation, respectively, compared to without irradiation, as demonstrated by staining H2kb-SIINFEKL complex. Immunofluorescence analysis confirmed this result (Fig. 6d). The highest H2kb-SIINFEKL complex signal (green) was observed on the surface of BMDCs that received TAzo-LNM/OVA complex treatment coupled with UV/Vis light irradiation. Moreover, we also observed that SAzo-LNM- and TAzo-LNM-treated groups showed significantly upregulated expression of the co-stimulatory markers CD80 and CD86 in BMDCs in the presence of UV/Vis irradiation, especially the latter, indicating enhanced DC maturation (Fig. 6e and Supplementary Fig. 13a). These results showed that light-irradiated LNMs could efficiently enhance tumour antigen cross-presentation in DCs, which is crucial to enhance T-cell-dependent antitumour response in vivo.

Interestingly, we found that nanomechanical action-induced endo-lysosomal rupture triggered NLRP3 inflammasome activation within BMDCs, which could also boost antitumour immunity. The activation of inflammasome were confirmed by WB immunoassay. As shown in Supplementary Fig. 13b and c, BMDCs treated with SAzo-LNMs and TAzo-LNMs exhibited higher NLRP3 expression after light irradiation, compared with DOTAP-LNPs and PBS. We then investigated the effects of LNMs on IL-1β release by enzyme-linked immunosorbent assay (ELISA), which was one hallmark of inflammasome formation. According to the results, SAzo-LNMs and TAzo-LNMs effectively increased the secretion of IL-1β, especially the latter, whereas DOTAP-LNPs had a moderate effect (Supplementary Fig. 13d). These results showed that light-irradiated LNMs could also activate the NLRP3−inflammasome pathway in BMDCs for cancer immunotherapy.

Since TAzo-LNM showed higher efficiency than SAzo-LNM in terms of inducing both antigen cross-presentation, DC maturation and inflammasome activation (Fig. 6c and e), we then chose TAzo-LNM as the vehicle for DC engineering and investigated DC-mediated antitumour activity in an animal model. B16F10-OVA (B16F10 cells express ovalbumin) tumour-bearing mice were injected subcutaneously with PBS and BMDCs, respectively (Fig. 6a, details in Supporting Information). BMDCs were pre-treated by TAzo-LNM/OVA complexes and received UV/Vis light irradiation (denoted as TAzo-LNM/OVA + UV/Vis). The cells without irradiation were employed as control (denoted as TAzo-LNM/OVA). As shown in Fig. 6f–h and Supplementary Fig. 14b, BMDCs that were pre-treated with TAzo-LNM/OVA + UV/Vis significantly delayed the growth of B16F10-OVA tumours and extended survival time of the mice compared to other treatment groups. Similar therapeutic tendencies were observed in the tumour slices by hematoxylin and eosin (H&E) staining and cell proliferation antigen Ki-67 staining (Supplementary Fig. 14a).

Then, the activation of cytotoxic CD8⁺ T cells within tumour tissues was analysed since CD8⁺ T cells are the major effectors in antitumour immunity. According to the results, the highest CD8⁺ T cell infiltration was found in the tumours collected from TAzo-LNM/OVA + UV/Vis group (CD8⁺, 23.4%), which should be attributed to the enhanced activation of BMDCs (Fig. 6i, and Supplementary Figs. 15a and 16a). Moreover, CD8⁺ T cells in the TAzo-LNM/OVA + UV/Vis group exhibited significantly increased expression of granzyme B (GzmB^high, 31.3%), confirming robust cytotoxic T cell activation (Fig. 6j and Supplementary Fig. 16b). Next, we studied the antitumour immune memory induced by this DC vaccine-based immunotherapy. As shown in Fig. 6k, and Supplementary Figs. 15b and 16d, TAzo-LNM/OVA + UV/Vis-treatment resulted in the elevation in the populations of CD62L^low CD44^high effector memory T cells (T_EM, 22.8%) and CD62L^high CD44^high central memory T cells (T_CM, 21.1%) in the spleens. Memory T cells are crucial for inhibiting tumour metastasis. To further confirm the antitumour immune memory, B16F10-OVA tumour-bearing mice in TAzo-LNM/OVA and TAzo-LNM/OVA + UV/Vis groups were rechallenged with B16F10-OVA cells (1 × 10⁶ cells per mouse) via intravenous (IV) injection on day 11. A PBS control group was also treated with B16F10-OVA cells. According

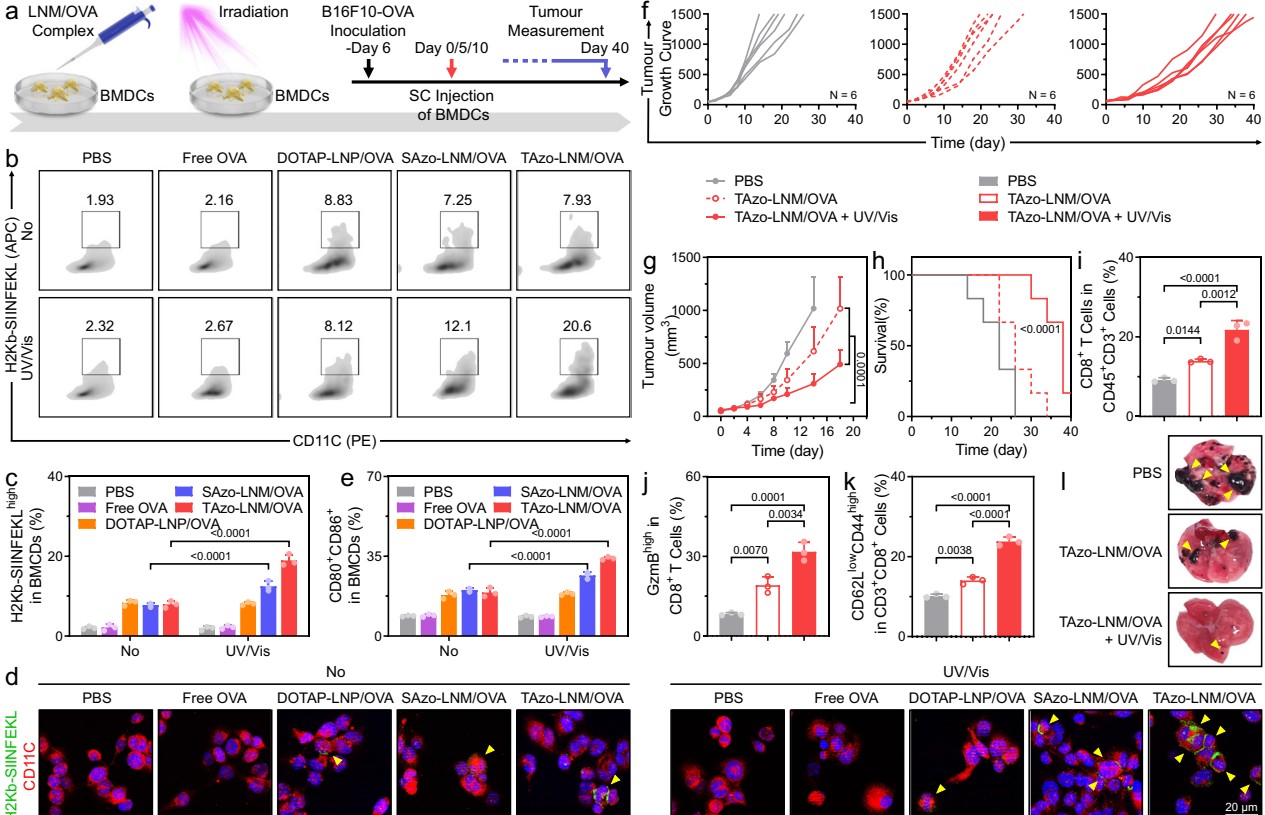

**Fig. 6 | LNM transports tumour antigens from endo-lysosomal compartments to cytoplasm for DC-based immunotherapy. a** Schematic illustration of the experimental design. The bone marrow-derived dendritic cells (BMDCs) are incubated with LNM/OVA complexes for 6 h, then received UV/Vis light irradiation for 10 min. After further incubation, the BMDCs are collected. B16F10-OVA tumour-bearing C57/BL6 mice are employed and SC injected BMDCs (1 × 10⁶ for per mouse). **b** Representative flow cytometry analysis and **c** Statistic analysis of the expression of H2kb-SIINFEKL in BMDCs after different treatments in vitro. **d** Fluorescence images of the expression of H2kb-SIINFEKL complex (green) on BMDCs (red, staining with anti-CD11c antibodies). **e** Flow cytometry analysis of the population of CD80⁺CD86⁺ BMDCs. **f** Individual tumour growth curves after different treatments. **g** Average tumour growth kinetics. **h** Survival of B16F10-OVA tumour-bearing C57/

BL6 mice. **i** Flow cytometry analysis of the population of CD8⁺ T cells (gated on CD45⁺CD3⁺ cells) in tumour tissues after different treatments. **j** Flow cytometry analysis of the expression of GzmB^high in CD8⁺ T cells (gated on CD45⁺CD3⁺ cells) in tumour tissues. **k** Flow cytometry analysis of the CD62L^low CD44^high T cells (gated on CD3⁺CD8⁺ cells) in the spleens. **l** Lungs collected from the mice after rechalleng-ing with intravenous injection of B16F10-OVA tumour cells (1 × 10⁶ cells per mouse). Data are presented as mean ± s.d. from *n* independent experiments (**c**, **e**, **i**, **j**, and **k**, *n* = 3; **f**, **g**, and **h**, *n* = 6). Statistical significance was analysed by two-way ANOVA with Sidak's multiple comparisons test for **c**, **e**, and **g**, log-rank (Mantel−Cox) test for **h**, one-way ANOVA with Tukey's multiple comparisons test for **i**, **j**, and **k**. *P* values are indicated.

to the results, treatment with TAzo-LNM/OVA + UV/Vis significantly inhibited tumour metastasis to the lungs (Fig. 6l). In contrast, the mice in other groups showed obvious lung metastasis. These results suggested that the LNM-based biomimetic strategy could enhance DC vaccine-based immunotherapy by directly regulating the transport of tumour antigens in DCs.

## Discussion

Inspired by intracellular molecular machines, we developed an artificial lipid-based molecular machine (LNM) composed of photo-isomerable Azo-based lipidoids and helper lipids, which achieved enhanced cytosolic transport of exogenous biologics and certain proteins. Unlike biological molecular machines, LNM was designed to perform mechanical movements by consuming photons. Reversible isomerization of Azo-based lipidoids, caused by simultaneous UV and Vis irradiation, enabled them to produce continuous rotation−inver-sion movements, accompanied by stretch−shrink motions. After entering cells, LNMs interacted with the endo-lysosomal membranes, and Azo-based lipidoids acted as rotors when they received UV/Vis light irradiation, resulting in enhanced destabilization and disruption of the membranes. Thus, the cargo could be effectively transported into the cytoplasm. In this sense, LNMs have similar functions to the

biological molecular machines that can regulate intracellular cargo transport in an ideal manner. Furthermore, LNMs have a similar structure and preparation procedure to clinically used LNPs, which makes this strategy applicable to a wide range of therapeutic nucleic acids and proteins.

In this work, we synthesized two different Azo-based lipidoids including SAzo and TAzo lipidoids to construct LNMs. We found that TAzo-LNMs exhibited higher transmembrane transport efficiency, although both were excellent. This is because compared to the SAzo lipidoid, a larger change in the molecular dimensions of the TAzo lipidoid takes place during isomerization. The higher transmembrane efficiency of TAzo-LNMs should be the result of these dramatic chan-ges in molecular dimensions. We demonstrated the capabilities of TAzo-LNMs to enhance the transfection efficiency of mRNA both in vitro and in vivo, as well as the Cre-mediated gene recombination at the cellular level. Moreover, we also showed the great potential of TAzo-LNMs in enhancing DC vaccine-based immunotherapy. TAzo-LNMs could carry the tumour antigens from endo-lysosomal compartments to the cytoplasm to bind with MHC-I in DCs when receiving light irradiation, resulting in robust cross-presentation. As a result, administration of the DCs that pre-treated with TAzo-LNM/tumour antigen complexes coupled with UV/Vis irradiation

significantly inhibited the growth of tumours and lung metastasis. Using the autologous dendritic cells (DCs) loaded with engineered tumour antigens (Provenge®) was the first FDA-approved cancer vaccine for the treatment of prostate cancer. Hence using LNM for DC engineering has clinical translation potential. However, we should also acknowledge some potential limitations of the LNM-based approach. For example, UV and Vis light have limited tissue penetration, resulting in an inability to manipulate LNMs located deep in tissue. Development of infrared light-driven LNMs is also expected to address the above-mentioned issue.

Our current efforts to control motions at the molecular levels may appear awkward compared with the exquisite functionalities displayed by natural systems. However, it should not be forgotten that the molecular machines produced by nature are complicated systems that have evolved over time. Our synthetic systems may not be as complex as these natural machines. Despite all this, our research still provides a viable route to fabricate man-made molecular machines with biological functions via simple chemistry, which has great value in constructing soft robotics, as well as designing intelligent (next-generation) drug delivery platforms.

# Methods

## Materials
Unless otherwise noted, reagents and solvents were used as received from commercial sources without further purification. Solvents were purchased from Sigma-Aldrich. Oxone® (Oxone|r, monopersulfate), *p*-Toluidine, *p*-phenylenediamine, dodecanoic acid, thionyl chloride, trimethylamine, triethylamine, N-bromosuccinimide, benzoyl peroxide, N,N'-dimethylethylenediamine (DMED), iodomethane, cholesterol (Chol), D-Luciferin potassium salt, and Cell Counting Kit-8 (CCK-8) were purchased from Thermo Scientific™. 1,2-Dioleoyl-sn-glycero-3-phosphoethanolamine (DOPE), 1,2-dioleoyl-3-trimethylammoniumpropane (DOTAP), and 1,2-dimyristoyl-rac-glycero-3-methoxypolyethylene glycol-2000 (DMG-PEG) were purchased from Avanti Polar Lipids. Carbon support film 5–6 nm thick on Square 200 mesh Copper Grid was purchased from Electron Microscopy Sciences. Fluorescein isothiocyanate (FITC) and rhodamine B isothiocyanate (RhB) were purchased from Sigma-Aldrich. Bicinchoninic acid disodium salt hydrate (BCA) was purchased from Solarbio. Dulbecco's modified Eagle medium (DMEM) growth medium, foetal bovine serum (FBS), and Dulbecco's phosphate buffered saline solution (D-PBS, 10 mM, pH 7.2) were purchased from Gibco (Gibco Corporation). Trypsin-EDTA (0.25%), LysoTracker Red, and LysoSensor Green were purchased from Invitrogen. The Hela (CCL-2) and B16F10 (CRL-6475) cell lines were purchased from the American-type culture collection (ATCC). Luciferase mRNA was purchased from Trilink BioTechnologies (San Diego, CA, USA). Peptide HiBiT (VSGWRLFKKIS), LgBiT expression vector, and Nano-Glo HiBiT Lytic Detection System were purchased from Promega (USA). The antibodies for western blot analysis, flow cytometry, and confocal laser scanning microscope observation (CLSM) were purchased from Invitrogen (Thermo Fisher Scientific), Cell Signaling Technology (USA), and Biolegend (USA).

## Preparation of LNM formulations
Preparation of SAzo-LNMs and TAzo-LNMs. Two different LNMs were prepared by co-assembling of SAzo or TAzo lipidoids with 1,2-dioleoyl-sn-glycero-3-phosphoethanolamine (DOPE), cholesterol (Chol), and 1,2-dimyristoyl-rac-glycero-3-methoxypolyethylene glycol-2000 (DMG-PEG) (molar ratio of 30:30:36.5:3.5, Azo units as standard) using the ethanol dilution method.

Preparation of mRNA-loaded LNMs. All lipids with specified molar ratios were dissolved in ethanol and mRNA was dissolved in 10 mM citrate buffer (pH 4.0). The two solutions were rapidly mixed at an aqueous-to-ethanol ratio of 3/1 by volume (3/1, aq./ethanol) to satisfy a final weight ratio of 50/1 (total lipids/mRNA, N/P molar ratio of 2.5),

then incubated for 10 min at room temperature and dialyzed at 4 °C against PBS.

## The mechanism of cytosolic transport
For flow cytometry assays, HeLa cells ($5 \times 10^4$ cells per well) were seeded in a 12-well plate and incubated overnight for cell attachment. 50 μg of different formulations were incubated with cells for 1 h, then received irradiation for 10 min. After another 30 min incubation, the cells were washed with PBS three times and stained with LysoSensor Green, followed by digesting using 0.25% trypsin for flow cytometry analysis.

For western blots, HeLa cells ($1 \times 10^5$ cells per well) were seeded in a six-well plate and incubated overnight for cell attachment. 50 μg of different formulations were incubated with cells for 1 h (and 4 h), then received irradiation for 10 min. After another 30 min incubation, the cells were washed with PBS three times and lysed in 0.5 ml of lysis buffer (PBS containing 1% Triton X-100 and 1 mM PMSF) at 4 °C for 30 min. Equal amounts of protein loadings were separated by SDS−PAGE and electrophoretically transferred to a PVDF membrane. Nonspecific binding sites were blocked with 5% nonfat milk in PBS and 0.05% Tween 20 (PBST) for 2 h at room temperature. Membranes were incubated in the primary antibodies (e.g., anti-mouse LAMP2 antibody (Invitrogen™), anti-mouse EEA1 antibody (Invitrogen™), anti-mouse Cath-D antibody (Proteintech Group Inc), anti-mouse GAPDH antibody (Novus Biologicals)) at appropriate concentrations for 24 h at 4 °C overnight, followed by three times rinse with PBST buffer for 15 min. The membranes were then incubated in the secondary antibodies (1:10,000) for 60 min at room temperature. The membranes were detected and analysed using a computerized chemiluminescent imaging system. Uncropped blots were presented in Supplementary Fig. 17.

## SAzo-LNMs and TAzo-LNMs facilitate cytosolic transport of mRNA
Confocal laser scanning microscope (CLSM) observation and flow cytometry were used to study the capability of SAzo-LNMs and TAzo-LNMs to reach the cytoplasm from endo-lysosomal compartments in HeLa cells. Different NBD-labelled formulations were prepared by using L-α-phosphatidylethanolamine-N-(4-nitro-benzo-2-oxa-1,3-diazole) (PE-NBD, a fluorescent lipid).

For CLSM observation (fluorescence imaging), HeLa cells were seeded in confocal dishes and allowed to grow until ~80% confluence. 50 μg of NBD-labelled different formulations (green) was added and cultured with cells for 1 h. The cells were then received UV/Vis light irradiation for 10 min. After another 30 min incubation, the cells were washed with PBS three times, and the endo-lysosomal compartments were stained with LysoTracker Red. Then, the cells were fixed using 4% paraformaldehyde, and the nucleus was stained with DAPI (blue).

For flow cytometry assays, HeLa cells ($5 \times 10^4$ cells per well) were seeded in a 12-well plate and incubated overnight for cell attachment. Different mRNA-loaded formulations (GFP mRNA, 1 μg mL$^{-1}$) were incubated with cells for 4 h and then received irradiation for 10 min. After another 20-h incubation, the cells were washed with PBS three times, followed by digesting using 0.25% trypsin for flow cytometry analysis.

## SAzo-LNMs and TAzo-LNMs facilitate cytosolic transport of mRNA in vivo
All animal procedures were performed with ethical compliance and approval by the Institutional Animal Care and Use Committee at Tufts University (Animal Protocol, M2020-51). The female BALB/c mice at 6−8 weeks old (Charles River) were used in this study. BALB/c mice were irradiated with UV/Vis light for 15 min at 1 hour and 2 h after subcutaneous (SC) injection of Luc mRNA-loaded LNMs (0.25 mg kg$^{-1}$ Luc mRNA, 5 μg per mouse), respectively. The power densities for 365 and >400 nm are 30 and 120 mW/cm$^2$, respectively. IVIS images were taken at 5 h post-injection.

### SAzo-LNMs and TAzo-LNMs facilitate cytosolic transport of Cre proteins

CLSM observation and flow cytometry were used to study the capability of SAzo-LNMs and TAzo-LNMs to transport Cre proteins from endo-lysosomal compartments to the cytoplasm. (−30)GFP-Cre (obtained by fusing a negatively charged GFP variant, (−30)GFP, to Cre), HeLa cells, and HeLa-DsRed cells were used in this study.

For CLSM observation (fluorescence imaging), HeLa cells were seeded in confocal dishes and allowed to grow until ~80% confluence. DOTAP-LNP/(−30)GFP-Cre, SAzo-LNM/(−30)GFP-Cre, and TAzo-LNM/(−30)GFP-Cre (green, protein, 1.5 µg mL$^{-1}$; formulation, 50 µg mL$^{-1}$) were added and cultured with cells for 1 h. The cells were then received UV/Vis light irradiation for 10 min. After further incubation, the cells were washed with PBS three times, and the endo-lysosomal compartments were stained with LysoTracker Red. Then, the cells were fixed using 4% paraformaldehyde, and the nucleus was stained with DAPI (blue).

For flow cytometry assays, HeLa-DsRed cells ($5 \times 10^4$ cells per well) were seeded in a 12-well plate and incubated overnight for cell attachment. DOTAP-LNP/(−30)GFP-Cre, SAzo-LNM/(−30)GFP-Cre, and TAzo-LNM/(−30)GFP-Cre (1.5 µg mL$^{-1}$ protein, green) were incubated with cells for 4 h, then received irradiation for 10 min. After another 20-h incubation, the cells were washed with PBS three times, followed by digesting using 0.25% trypsin for flow cytometry analysis.

### Antitumour efficacy in a mouse model of melanoma

The female C57BL/6 mice at 6–8 weeks old (Charles River) were housed in a specific pathogen-free-grade animal facility with an air humidity of 40–70%, ambient temperature ($22 \pm 2\,^{\circ}C$), and 12-h dark/12-h light cycle. B16F10 tumour cells expressing ovalbumin (B16F10-OVA) were used in this study. B16F10-OVA tumour-bearing mice were randomly divided into three groups ($n = 6$) after 6 days of inoculation. The mice were then subcutaneously injected with BMDCs ($1 \times 10^6$ per mouse) every 5 days three times. The BMDCs were pre-treated by TAzo-LNM/OVA complexes with and without UV/Vis light irradiation. The tumour volume ($V$) was calculated according to the formula: $V = L \times W2/2$, where $L$ and $W$ were the longest and shortest diameter (mm) of the tumour, respectively. The mice were euthanized when exhibiting signs of impaired health or when the volume of the tumour exceeded 1.5 cm$^3$ (Humane Endpoints).

For flow cytometry assays, freshly harvested tumour tissues were digested with collagenase IV and made into single-cell suspensions according to the manufacturer's instructions. After that, cells were collected and diluted to $1 \times 10^7$ cells mL$^{-1}$. 100 µL cells were stained by adding a cocktail of fluorescent conjugated antibodies on the ice for flow cytometry analysis (e.g., APC-anti-mouse OVA257-264 (SIINFEKL) peptide bound to H-2Kb antibody (eBioscience™), FITC anti-mouse CD80 antibody (BD Biosciences), PE anti-mouse CD80 antibody (BD Biosciences), APC/Cy7 anti-mouse CD45 antibody (BioLegend), APC anti-CD3 antibody (BD Biosciences), PE anti-mouse CD8 antibody (BD Biosciences), FITC-anti-mouse CD4 antibody (BD Biosciences), and FITC-anti-mouse GzmB antibody (BioLegend)).

### Statistical analysis

Data were expressed as mean ± SD. All data were analysed using Flowjo V10 and GraphPad Prism (8.0) software.

### Reporting summary

Further information on research design is available in the Nature Portfolio Reporting Summary linked to this article.

## Data availability

All data supporting the findings of this study are available within the article and its supplementary files. Any additional requests for information can be directed to, and will be fulfilled by, the corresponding author. Source data are provided with this paper.

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

## Acknowledgements

We acknowledge the financial support by the NIH grants R01 EB027170.

## Author contributions

Y.Z. and Q.B.X. devised the project. Y.Z. synthesized the lipidoid compounds. Y.Z. carried out the experimental work and analysed the data. Z.F.Y. and D.H.S. participated in the discussion of the results. S.L.G. provided the TEM images. D.W. and J.K. polished the language. Y.Z. and Q.B.X. wrote the manuscript.

## Competing interests
Q.X. and Y.Z. are inventors of a pending patent related to this work filed by Tufts University (Application number: 63430797). Q.X. is the founder and consultant of Hopewell Therapeutics Inc. The remaining authors declare no competing interests.
