## [Peer Review File · Nature Communications]

REVIEWER COMMENTS

Reviewer #1 (Remarks to the Author):

This manuscript by Zhao et al. reports a novel approach to enhancing lipid nanoparticles to escape endosomes/lysosomes through co-encapsulation of azobenzene (Azo) lipidoids, which are designed to disrupt endosomes/lysosomes following UV/Vis irradiation. The authors demonstrated that the approach allows for enhanced delivery of various payloads, including mRNA and proteins, for biological applications. This is an interesting study which is well designed and presented. There are a few concerns.

1. The major hypothesis regarding the delivery mechanism is that Azo lipidoids within LNMs directly disrupt endo-lysosomal membrane through rotation-invasion, resulting in enhanced cytosolic transport. However, there could be an alternative explanation that rotation-invasion of Azo lipidoids disrupts LNMs, leading to exposure of amino groups of Azo lipidoids, which subsequently disrupt endosomes/lysosomes through a proton sponge effect. TEM images in Fig. 11c, which show some LNMs are not in spherical shape after UV/Vis irradiation, support the latter hypothesis.
2. To better determine the mechanism, further characterization of LNMs is needed. Would UV/Vis irradiation alter the size of LNMs, release encapsulated payloads, and expose amino groups in Azo lipidoids?
3. In Fig. 2b&c, were SAzo and TAzo lipidoids tested at the same concentration? Should TAzo lipidoids have higher absorbance than SAzo if testing at the same concentration? Similarly, in Fig. 2f&g, were SAzo-LNM and TAzo-LNM tested at the same concentration? It is shown that the absorbances for both LNMs are similar.
4. Fig. 3a characterized the cytotoxicity, but not biocompatibility as described.
5. Data in Figures 3-5 show that TAzo- LNM is more efficient than SAzo-LNM in delivery of mRNA/protein. The authors attributed this observation to the greater change in molecular dimensions of TAzo-LNMs. Compared to SAzo-LNM, TAzo-LNMs have more (2 times) amino groups. One cannot exclude the possibility that the key factor is not the change in molecular dimensions but the amount of amino groups.
6. UV/Vis light has limited tissue penetration. To better prepare for potential biological applications, the authors need to characterize of the effect of UV/Vis irradiation on mRNA/protein delivery in deep tissue, for example, by analyzing the spatial distribution of luciferase expression in tumor tissues in Fig. 4h.

Reviewer #2 (Remarks to the Author):

The manuscript “Nanomechanical Action Opens Endo–Lysosomal Compartments” by Zhao et al. reports the fabrication of azobenzene (Azo) lipidoids co-assembled lipid-based nanomachine (LNM), with which the reversible isomerization of Azo groups may enhance endo–lysosomal transport (escape) and facilitate efficient delivery of various biologics into the cytoplasm. Considering that the azobenzene (Azo) lipidoids co-assembled lipid-based nanomachine has been reported in author’s previous works, making the material here not new, the key innovation lies in the nanomechanical action induced opening of endo–lysosomal. This part is, however, lacking some solid proves and details. Therefore, I suggest that the author revise the manuscript to delve deeper into the mechanisms and processes of the relevant system.

(1) During the assembly process, are the azobenzene groups containing phospholipid molecules in the cis or trans configuration? Will the cis/trans structure affect the morphology of the liposomes formed?

(2) Dynamic molecules exist in the structure, is there a more intuitive characterization method? Currently, it is characterized by UV absorption, which is not very intuitive.

(3) Indeed, the author has demonstrated the necessity of azobenzene phospholipid molecules for liposomes to escape from lysosomes, but there is still a lack of critical evidence for the specific escape method:

1. How to prove that it is the azobenzene phospholipid molecule exchanged into the lysosomal membrane that leads to lysosomal rupture, rather than other factors (such as the vibration, deformation, and rupture of liposomes)?

2. Azobenzene-containing liposomes have higher delivery efficiency than azobenzene-free liposomes. How to exclude the interference caused by structural changes of the azobenzene functional group on the release of guest substances from liposomes?

3. Did the liposomes escape from the lysosomes and then break to release the guest substances, or did the liposomes first break to release the guest substances into the lysosomes and then the broken fragments were inserted into the lysosomes, further leading to lysosomal rupture in the cis/trans isomerization of azobenzene?

4. Can we directly observe the contact, exchange, and lysosomal rupture of liposomes and lysosomal membranes through slicing/cryo-electron microscopy?

(4) In theory, it is unlikely that all liposomes in the system are located in lysosomes when irradiated with UV light. Will some liposomes that are located outside the cells or are being phagocytized cause changes in cell uptake due to mechanical forces, resulting in bypassing lysosomes?

Reviewer #3 (Remarks to the Author):

The submitted manuscript by Zhao et al., describes the use of lipidoids to improve Endo-lysosomal-cytosol delivery of biologics. The hypothesis is that by increasing endosomal destabilization phagocytosed proteins or nucleic acid, given in a therapeutical setting, more easily can reach the cytosol.

In the first series of experiments the authors describe the process of photoisomerization of the SAzo and TAzo lipidoids following UV irradiation; the cis-to-trans and trans-to-cis process allow a rotation-inversion and stretch-shrink required to destabilize the endosomal membranes (Figure 2 and 3). The authors go on in determining that this process facilitate endosomal escape and cytosolic transport of mRNA. The experiments are performed in both cultured cells as well as in mice, subcutaneously injected with SAzo and TAzo lipidoids and fluorescent RNA (Figure 4). Similar experiments are then performed on fluorescently labeled protein to be delivered for genome editing (Figure 5). At last, the authors describe the use of this system to facilitate cross-presentation of endosomal delivered MHC-I antigens (Figure 6).

There are few important issues to be considered.

1) albeit the authors describe a "positive" potential role of endosomal destabilization, in reality in many human diseases endosomal destabilization has been shown to be a great trigger for innate immune responses. To name few, in aging lipofuscin aggregates have been shown to embed in the endosomal membrane and facilitate proton escape and inflammasome activation. A similar role has been shown for molecular debris and uric acid crystals in gout. As such the authors should at least show whether the "therapeutical" endo-cytosol molecular delivery is associated with NLRP inflammasome activation and IL1 secretion.

2) As a control the author should analyze for the presence of "active" Cathepsins in the cytosol. If RNA and proteins "therapeutically" escape the endosomes so should endosomal proteases, which could induce cellular necrosis.

3) How long is the endosomal destabilization last? This is very important since the therapeutical edge could be lost by the ensue inflammatory process due to the cytosolic escape of cathepsins and other endosomal proteases, as well as H⁺ ions into the cytosol. Time course experiments should be included, as well as the lipidoids half-life.

4) The need for UV/Vis light irradiation needs to be better conceptualized in terms of human therapy, since the authors propose to use the lipidoids in molecular delivery. In mice they propose total body UV irradiation what about in human? In skin cancer it is conceivable to localize the UV irradiation to a defined area but what about solid tumors of parenchymal organs? Also, would the UV light penetrate deep enough to achieve the required SAzo and TAzo photoisomerization in parenchymal organs?

5) There is also a novelty issue with the manuscript, since the use of lipidoids has been extensively reported in the literature.

Reviewer #4 (Remarks to the Author):

Zhao Y et al., demonstrated with numerous experiments, the workable model of an artificial lipid based molecules composed of photoisomerable Azo-based compounds to achieve the enhanced transport of exogenous biologics or targeted drugs. The experiments are performed meticulously, but there are several concerns that must be addressed.

1) There is a lot explored about the benefits and efficiency of using azobenzene products as a cancer therapeutics and advantages of phototriggered delivery systems, please include the references and discuss them in the introduction.

2) The experiments performed by the authors to show the nano-mechanical action of the SAzo and TAzo lipidoids which opens endo-lysosomal compartments are amplified significantly in presence of UV light. The significance of phototherapy alone or in combination with certain drugs in treating certain tumors is well studied. It will be good to see the effect of the UV alone on the tumor reduction as a control. It is an important control which is missing in all the experiments.

3) The effect of phototherapy is in clinics for treating skin cancers. The authors have studied the effect of the lipidoids in presence of UV light in the melanoma mouse model. Again, the control of the UV light only is missing in these experiments. Also, it will be good to explore other cancer models.

In the experiments showing the antitumor efficacy in the mouse model of melanoma, it will be worthwhile to introduce the SAzo and TAzo lipidoids at the site of the melanoma and use the site-specific UV radiation and see if the tumor gets reduced including all the appropriate controls. The model system where BDMCs are irradiated and then introduced to see the effect on tumor or irradiating the entire mouse cannot be translated in the patients, hence lacks bench to bed side essential perspective.

4) In fig 4g, where the mice are irradiated after the SC injection, it will be worthwhile to investigate if other non-specific effects are seen on the other cellular compartments or other organs in the invivo conditions.

5) Please include the experiments and details on the stability and the kinetics (half-life) of the SAzo and TAzo lipidoids inside the cell?

6) Considering that the lipidoids, azobenzene products and the lipid-based drug delivery has been studied and explored by different groups, please discuss the novelty of the study which is questionable where it stands now.

7) The translational significance and shortcomings of the approach should be also discussed.

Reviewer(s)' Comments to Author:

Reviewer #1

This is an interesting study which is well designed and presented. There are a few concerns.

1. The major hypothesis regarding the delivery mechanism is that Azo lipidoids within LNMs directly disrupt endo-lysosomal membrane through rotation-invasion, resulting in enhanced cytosolic transport. However, there could be an alternative explanation that rotation-invasion of Azo lipidoids disrupts LNMs, leading to exposure of amino groups of Azo lipidoids, which subsequently disrupt endosomes/lysosomes through a proton sponge effect. TEM images in Fig. 11c, which show some LNMs are not in spherical shape after UV/Vis irradiation, support the latter hypothesis.

Response: Thank you for your comments. This is an interesting point, however, in our system the Azo-based lipidoids contain a quaternary ammonium cation and their chemical structures are shown in **Fig. 1a**. Azo-based lipidoids cannot be further protonated, hence the mechanism for endo/lysosomal escape in our system is not based on the proton sponge effect.

We believe that the endosomal escape mechanism of quaternary ammonium-based formulations is based on endosome destabilization through direct interaction between the ammonium cations and endosomal membranes (*ACS Nano*, **2013**, *7*, 3767.).

2. To better determine the mechanism, further characterization of LNMs is needed. Would UV/Vis irradiation alter the size of LNMs, release encapsulated payloads, and expose amino groups in Azo lipidoids?

Response: Thank you for your suggestions. We have performed additional experiments to evaluate both the size change of LNMs and the payload release after UV/Vis irradiation.

The size changes of SAzo-LNMs and TAzo-LNMs after UV/Vis irradiation were evaluated using DLS. As shown in **Fig. S11d** and **S11e**, UV/Vis irradiation did not cause great changes in the particle size of SAzo-LNMs and TAzo-LNMs, but their PDI values increased after irradiation. The increase in PDI values might be attributed to the loose nanostructure of LNMs caused by nanomechanical action, which is consistent with TEM results.

To assess whether UV/Vis irradiation disrupts the LNM and releases the payload (mRNA), we monitored the release of mRNA from LNMs with and without light irradiation using agarose gel electrophoresis. Free mRNA was used as a control. As demonstrated in **Fig. S10c**, almost all mRNA was retained near the gel wells with or without UV/Vis irradiation in both SAzo-LNM and TAzo-LNM groups, indicating that no free mRNA was released from LNMs regardless of light irradiation. Interestingly, we observed a slight band migration of UV/Vis irradiated LNMs compared to non-irradiated samples. This slight band migration showed that the light irradiation does change the conformation or nanostructure of LNMs but does not cause disruption. The bands from the light-irradiated samples are still very sharp (no smearing), indicating the mRNA/LNM samples are still relatively homogeneous. We have added a

paragraph of discussion of these results in main text (Page 9) and Supporting Information (Page S18).

Revised Supplementary Figure S11. d) The changes of particle size of DOTAP-LNP, SAzo-LNM, and TAZo-LNM with and without UV/Vis irradiation. e) The changes of PDI values of DOTAP-LNP, SAzo-LNM, and TAZo-LNM with and without UV/Vis irradiation.

Revised Supplementary Figure S10. c) Agarose gel electrophoresis of GFP mRNA-loaded different LNMs with and without UV/Vis irradiation.

3. In Fig. 2b&c, were SAzo and TAZo lipidoids tested at the same concentration? Should TAZo lipidoids have higher absorbance than SAzo if testing at the same concentration? Similarly, in Fig. 2f&g, were SAzo-LNM and TAZo-LNM tested at the same concentration? It is shown that the absorbances for both LNMs are similar.

Response: Thank you for your comments. In **Fig. 2b** and **2c**, the concentrations of SAzo and TAZo lipidoids are different. The SAzo lipidoid was prepared as 30 μ M solution in anhydrous dimethyl sulfoxide (DMSO) while the TAZo lipidoids was prepared as 15 μ M solution in DMSO (**Section 4** in Supporting Information). Considering that each TAZo lipidoids contains two Azo units, the number of Azo units in SAzo lipidoids and TAZo lipidoids samples should be equal.

In **Fig. 2f** and **2g**, we prepared two different LNMs using SAzo or TAZo lipidoids, DOPE, cholesterol (Chol), and DMG-PEG with the following molar ratios.

SAzo-LNM: the molar ratio of SAzo lipidoids:DOPE:Chol:DMG-PEG was 30:30:36.5:3.5.

TAzo-LNM: the molar ratio of TAZo lipidoids:DOPE:Chol:DMG-PEG was 15:30:36.5:3.5 (30:30:36.5:3.5, Azo units as standard).

The number of Azo units in SAzo-LNM is equal to that in TAZo-LNM. To clarify this, we moved the formulation information from the Supporting Information to the main text.

4. Fig. 3a characterized the cytotoxicity, but not biocompatibility as described.

Response: Thank you for your suggestions. We have corrected the discussion about LNM cytotoxicity to avoid confusion.

5. Data in Figures 3-5 show that TAzo- LNM is more efficient than SAzo-LNM in delivery of mRNA/protein. The authors attributed this observation to the greater change in molecular dimensions of TAzo-LNMs. Compared to SAzo-LNM, TAzo-LNMs have more (2 times) amino groups. One cannot exclude the possibility that the key factor is not the change in molecular dimensions but the amount of amino groups.

Response: Thank you for your comments. As shown in the response to question #3, the number of Azo units and the number of quaternary ammonium groups in SAzo-LNMs and TAzo-LNMs are the same.

6. UV/Vis light has limited tissue penetration. To better prepare for potential biological applications, the authors need to characterize of the effect of UV/Vis irradiation on mRNA/protein delivery in deep tissue, for example, by analyzing the spatial distribution of luciferase expression in tumour tissues in Fig. 4h.

Response: We agree with the reviewer that UV/Vis light has limited tissue penetration. In this manuscript, we aim to evaluate a novel concept to open endo-lysosomes through nanomechanical action. It's been shown in the literature that other approaches such as using NIR combined with upconverting nanoparticles, can provide deeper tissue penetration (*J. Am. Chem. Soc.*, **2018**, 140, 50, 17656–17665; *Angew. Chem.*, **2019**, 131, 18375–18379.), but it is out of the scope of current research.

Reviewer #2

Key innovation lies in nanomechanical action-induced opening of endo-lysosomes. I suggest that the author revise the manuscript to delve deeper into the mechanisms and processes of the relevant system.

1. During the assembly process, are the azobenzene groups containing phospholipid molecules in the cis or trans configuration? Will the cis/trans structure affect the morphology of the liposomes formed?

Response: Thank you for your comments. During the assembly process, almost all of the azobenzene groups are in the *trans* configuration. The NMR characterization of our Azo-based lipidoids supports this statement. As shown in **Fig. S4a** and **S5b**, the chemical shift values of the protons in the benzene ring lie between 7.7 and 8.1 ppm, indicating that the azobenzene units are in the *trans* state. This is because *trans*-isomers are more stable than *cis*-isomers, which has been confirmed by many previous works (*J. Am. Chem. Soc.*, **2011**, 133, 49, 19684–19687; *Organometallics*, **2012**, 31(17): 6262–6269.).

In terms of TEM results, the *cis/trans* structure will affect the morphology of the LNMs. As demonstrated in **Fig. S11c**, light-induced *cis/trans* isomerization of Azo-based lipidoids loosened the nanostructure of LNMs.

2. Dynamic molecules exist in the structure, is there a more intuitive characterization method? Currently, it is characterized by UV absorption, which is not very intuitive.

Response: Thank you for your comments. It is very difficult to use existing characterization tools, while simultaneously introducing UV/Vis irradiation *in situ* to observe dynamic molecule configuration changes of the samples. However, considering that both *cis*-isomers and *trans*-isomers reach a thermodynamic equilibrium state via thermal relaxation at a much slower rate (>48 hours, **Fig. S6h** and **S6i**), Azo-based lipidoids could maintain a specific configuration for a period after removing the light that is long enough to be quantitatively analyzed using an existing tool such as ¹H NMR. An ¹H NMR spectrum could be useful to reflect the instantaneous percentages of *cis*- and *trans*-isomers during light irradiation. We irradiated the samples with different combinations of UV and Vis light (shown in **Fig. S6g**). We used a 180s exposure time which is longer than the time required to reach photostability states of azobenzene (usually 30s). Then, after the irradiation, the samples were immediately analyzed using ¹H NMR.

The chemical shift values of the protons in *trans*-azobenzene phenyl rings lie between 7.7 and 8.1 ppm, and the chemical shift values of the protons in *cis*-azobenzene phenyl rings lie between 6.8 and 7.6 ppm. As shown in **Fig. S6g**, the isomerization of the azobenzene upon light irradiation is fully revealed in the ¹H NMR. Specifically, increasing the UV light intensity resulted in a shift of the isomerization equilibrium towards the *cis*-rich state, whereas increasing the Vis light intensity shifted the equilibrium towards the *trans*-rich state. These results indicated that Azo-based lipidoids underwent dynamic isomerization during light exposure. We have added a paragraph of discussion of these results in main text (Page 5) and Supporting Information (Page S13).

Revised Supplementary Figure S6. g) ¹H NMR spectrum of Azo units in Azo-based lipidoids after simultaneous UV and Vis irradiation. h) and i) Thermal relaxation of different Azo-based lipidoids in dark.

3. Indeed, the author has demonstrated the necessity of azobenzene phospholipid molecules for liposomes to escape from lysosomes, but there is still a lack of critical evidence for the specific escape method:

(1) How to prove that it is the azobenzene phospholipid molecule exchanged into the lysosomal membrane that leads to lysosomal rupture, rather than other factors (such as the vibration, deformation, and rupture of liposomes)?

Response: Thank you for your comments. The general accepted mechanism of cationic based delivery systems is that cationic lipids fuse with endo-lysosomal membranes due to the electrostatic interaction between the lipids and membrane, leading to disruption of endo-lysosomal membrane (*Gene Therapy*, **2001**, 8, 1188–1196; *ACS Nano*, **2013**, 7, 3767.). In our system, the Azo-based lipidoids containing quaternary ammonium groups are cationic lipids, and the cytosolic transport mechanism of Azo-based lipidoids in the absence of UV/Vis irradiation would be similar to conventional cationic lipids. Introducing light irradiation could generate the cis/trans isomer transitions, leading to enhanced disruption of the endo-lysosomal membrane.

The exact mechanism of the enhanced endo-lysosomal disruption could be due to vibration, deformation, and rupture, or through the proposed nanomechanical action. We lean toward the nanomechanical action, because the samples were simultaneously irradiated using UV and Vis light and such treatment would create a continuous rotation of the Azo-based lipidoids rather

than a vibration or deformation.

Furthermore, the forces that cause deformation and rupture of endo-lysosomal membranes are relatively large and require very dramatic morphological changes in the nanoparticles. In a recent study by Liang and co-workers (*Nat. Nanotechnol.*, **2020**, 15(12), 1053–1064.), a proton-driven nanotransformer is reported. In this system, the escape mechanism is based on deformation of nanoparticles leading to rupture of the endo-lysosomal membrane. The nanotransformer needs to undergo a dramatic morphological change from nanospheres (about 100 nanometres in diameter) into nanosheets (several micrometres in length or width) to effectively disrupt the endosomal membrane. In our system, we did not observe drastic changes in LNM particle size (**Fig. S11d** and **S11e**) and morphology (**Fig. S11c**) upon light irradiation.

Revised Supplementary Figure 11. d) The changes of particle size of DOTAP-LNP, SAzo-LNM, and TAzo-LNM after UV/Vis irradiation. e) The changes of PDI values of DOTAP-LNP, SAzo-LNM, and TAzo-LNM after UV/Vis irradiation.

(2) Azobenzene-containing liposomes have higher delivery efficiency than azobenzene-free liposomes. How to exclude the interference caused by structural changes of the azobenzene functional group on the release of guest substances from liposomes?

Response: Thank you for your comments. As shown in **Fig. 4f** (also **Fig. 3c**), we didn't find dramatic differences between DOTAP LNP groups (azobenzene-free formulation) and SAzo/TAzo-LN groups in terms of cytosolic transport efficiency without light irradiation. These results demonstrate that adding the azobenzene functional group has limited effects on the guest substance release from LNMs.

(3) Did the liposomes escape from the lysosomes and then break to release the guest substances, or did the liposomes first break to release the guest substances into the lysosomes and then the broken fragments were inserted into the lysosomes, further leading to lysosomal rupture in the cis/trans isomerization of azobenzene?

Response: Thank you for your comments. As shown in **Fig. S10c**, we observed that the light irradiation doesn't cause the disruption of the LNM. Hence, we believe that light irradiation will not cause the premature disruption of the LNM in endo-lysosomes.

Revised Supplementary Figure S10. c) Agarose gel electrophoresis of GFP mRNA-loaded different LNMs with and without UV/Vis irradiation.

(4) Can we directly observe the contact, exchange, and lysosomal rupture of liposomes and lysosomal membranes through slicing/cryo-electron microscopy?

Response: Thank you for your suggestions. This would be a very informative experiment. However, it is technically very challenging. Unlike most of other studies with inorganic nanoparticles (such as gold, silica containing heavy elements), the LNMs consist largely of light atoms, and their composition is very similar to cellular components. Their elastic interactions with highly energetic electrons are relatively weak and it is hard to distinguish them inside the cell using cryo-electron microscopy. (*Polym. Rev.*, **2010**, 50 (3), 321–339.). Thus, in this work, we used confocal laser scanning microscopy (CLSM), western blotting immunoassay (WB), and the split luciferase endosomal escape quantification (SLEEQ) assay to assess nanomechanical action-mediated endo-lysosomal escape.

4. In theory, it is unlikely that all liposomes in the system are located in lysosomes when irradiated with UV light. Will some liposomes that are located outside the cells or are being phagocytized cause changes in cell uptake due to mechanical forces, resulting in bypassing lysosomes?

Response: Thank you for your comments. We agree with the reviewer that nanomechanical action-mediated membrane destabilization is a universal mechanism for membrane destabilization. In **Fig. 4c** and **4d**, we observed that the LNMs can enhance the destabilization of endosomes and lysosomes, respectively. Hence, it is very likely the LNMs can also destabilize the cell membrane upon light irradiation if the LNMs are adhered to the cell membranes.

Reviewer #3

There are few issues to be considered.

1. Albeit the authors describe a "positive" potential role of endosomal destabilization, in reality in many human diseases endosomal destabilization has been shown to be a great trigger for innate immune responses. To name few, in aging lipofuscin aggregates have been shown to embed in the endosomal membrane and facilitate proton escape and

inflammasome activation. A similar role has been shown for molecular debris and uric acid crystals in gout. As such the authors should at least show whether the “therapeutic” endo-cytosol molecular delivery is associated with NLRP3 inflammasome activation and IL1 secretion.

Response: Thanks for your valuable suggestions. It is reasonable that the endo-lysosomal disruption could lead to NLRP3 inflammasome activation. We have assessed the NLRP3 inflammasome activation and IL-1 β secretion at the cellular level using bone marrow-derived dendritic cells (BMDCs) treated with our LNMs. As shown in **Fig. S13b** and **S13c**, BMDCs treated with SAzo-LNMs and TAZo-LNMs exhibited higher NLRP3 expression, compared with DOTAP-LNPs and PBS. We also found that SAzo-LNMs and TAZo-LNMs effectively increased the secretion of IL-1 β , whereas DOTAP-LNPs had a moderate effect (**Fig. S13d**). These results indicate that light-irradiated LNMs could activate the NLRP3-inflammasome pathway through destabilizing the endo-lysosomal membranes. This finding is also consistent with the delivery efficiency results, and supports our conclusion that the increased delivery from LNMs (upon light irradiation) is due to increased endo-lysosomal escape. We have added a paragraph of discussion of these results in main text (Page 14) and Supporting Information (Page 20).

Revised Supplementary Figure 13. b) and c) Detection of NLRP3 expression in BMDCs after different treatments using western blotting analysis. d) IL-1 β secretion in BMDCs. BMDCs are incubated with different LNMs for 6 hours, then received UV/Vis light irradiation for 10 min. After another 18 h incubation, BMDCs and the culture medium were collected for western blotting immunoassay (WB) and enzyme-linked immunosorbent assay (ELISA).

2. As a control the author should analyze for the presence of “active” Cathepsins in the cytosol. If RNA and proteins “therapeutically” escape the endosomes so should endosomal proteases, which could induce cellular necrosis.

Response: Thank you for your suggestions. We agree that the endo-lysosomal disruption will lead to the release of cathepsins and other digestive enzymes, which can induce cellular necrosis. However, the cells typically have a degree of tolerability to digestive enzymes released from endo-lysosomes that do not cause necrosis. Most of the reported delivery systems shown in the literature have relatively low endosomal escape efficiency (<2%). In the LNMs, the efficiency increased to 3.6% and it seems the cells are still able to tolerate this increase. It would be interesting to see at what level of endo-lysosomal disruption would go beyond cell tolerability and cause massive necrosis. To increase the delivery efficiency (through the endosomal uptake mechanism), it is necessary to increase the efficiency of endo-lysosomal

escape. However, this problem is a double-edged sword, and we need to keep a balance between the level of endo-lysosomal disruption and delivery efficiency for any practical applications.

3. How long is the endosomal destabilization last? This is very important since the therapeutical edge could be lost by the ensue inflammatory process due to the cytosolic escape of cathepsins and other endosomal proteases, as well as H⁺ ions into the cytosol. Time course experiments should be included, as well as the lipidoids half-life.

Response: Thank you for your comments. As we discussed in Response to question #2, the amount of endo-lysosomal disruption is a double-edge sword. Too high or too low are both not desired. In our LNM-based delivery system, we can control the endo-lysosome disruption level by the duration of the light irradiation.

4. The need for UV/Vis light irradiation needs to be better conceptualized in terms of human therapy, since the authors propose to use the lipidoids in molecular delivery. In mice they propose total body UV irradiation what about in human? In skin cancer it is conceivable to localize the UV irradiation to a defined area but what about solid tumours of parenchymal organs? Also, would the UV light penetrate deep enough to achieve the required SAzo and Tazo photoisomerization in parenchymal organs?

Response: We agree with the reviewer that UV/Vis light has limited tissue penetration. The novelty of this work is to evaluate a concept of opening endo-lysosomes through nanomechanical action and verify its feasibility. For future effort to increase the tissue penetration, we may use other approaches such as a combination of NIR light and upconverting nanoparticles (UCNPs) to enable the photoisomeration of Azo lipidoids (*J. Am. Chem. Soc.*, **2018**, 140, 50, 17656–17665; *Angew. Chem.*, **2019**, 131, 18375–18379.).

5. There is also a novelty issue with the manuscript, since the use of lipidoids has been extensively reported in the literature.

Response: Thank you for your comments. Even though we and many other groups have extensively published works using lipidoids as a delivery system, using light irradiation to generate the nanomechanical action to enhance endo-lysosome disruption and delivery efficiency is a novel concept. The data shown in this manuscript demonstrated the feasibility of using the combination of chemical (charge interaction) and mechanical (light induced rotation) approaches to enhance the cytosolic delivery.

Reviewer #4

The experiments are performed meticulously, but there are several concerns that must be addressed.

1. There is a lot explored about the benefits and efficiency of using azobenzene products as a cancer therapeutics and advantages of phototriggered delivery systems, please include the references and discuss them in the introduction.

Response: Thank you for your suggestions. We have added a brief discussion of the advantages of photo-triggered delivery systems in the introduction. The references have also been added to the revised manuscript as shown in Page 2, as following:

Lipid-based nanomachine (LNM) is designed to convert light energy into mechanical movements (*Nano Res.*, **2018**, 11: 5424–5438; *Angew. Chem.*, **2022**, 134(30): e202116073.). This is because photons are one of the most convenient energy inputs to drive LNMs at a high spatiotemporal resolution. Azo-based lipidoids were synthesized to perform light-driven mechanical movement, in which its operation was based on an efficient and clean photoreaction, namely, photoisomerization (*Nature*, **2016**, 537, 179–184.).

2. The experiments performed by the authors to show the nano-mechanical action of the SAzo and TAzO lipidoids which opens endo-lysosomal compartments are amplified significantly in presence of UV light. The significance of phototherapy alone or in combination with certain drugs in treating certain tumours is well studied. It will be good to see the effect of the UV alone on the tumour reduction as a control. It is an important control which is missing in all the experiments.

Response: Thank you for your suggestions. Unlike traditional phototherapy that directly irradiates the tumour site, we did not irradiate the tumours using UV light. We just irradiated BMDCs *in vitro* to improve the efficiency of tumour antigen cross-presentation and DC activation. Then, we injected activated BMDCs into mice to activate a robust anti-tumour immune activity in a tumour-bearing mouse model.

Our preliminary experiments on GFP mRNA transfection showed that UV irradiation alone was not effective in increasing transfection efficiency when the cells were incubated with GFP mRNA-loaded LNMs (**Fig. S17**). These results indicated that a single UV irradiation (10 min) could not improve the cytosolic drug transport. This might be attributed to the unidirectional conformational change of Azo-based lipidoids triggered by UV irradiation, which could not effectively destabilize the endo-lysosomal membranes. That is to say, irradiation of Azo-based lipidoids with only UV light cannot produce effective nanomechanical movements. Thus, we did not use single UV irradiation in our study design, rather we focused on the UV/vis irradiation which can generate continuous rotation. These results also support the importance of the continuous rotation on the LNM-mediated endo-lysosomal disruption.

Supplementary Figure S17. GFP mRNA transfection efficacy of DOTAP-LNP, SAzo-LNM, TAzo-LNM ($1 \mu\text{g mL}^{-1}$, GFP mRNA) tested on HeLa cells by quantifying GFP-positive cells. (Data for reviewing only)

3. The effect of phototherapy is in clinics for treating skin cancers. The authors have studied the effect of the lipidoids in presence of UV light in the melanoma mouse model. Again, the control of the UV light only is missing in these experiments. Also, it will be good to explore other cancer models.

In the experiments showing the antitumour efficacy in the mouse model of melanoma, it will be worthwhile to introduce the SAzo and TAzo lipidoids at the site of the melanoma and use the site-specific UV radiation and see if the tumour gets reduced including all the appropriate controls. The model system where BMDCs are irradiated and then introduced to see the effect on tumour or irradiating the entire mouse cannot be translated in the patients, hence lacks bench to bed side essential perspective.

Response: Thank you for your suggestions. Our purpose is not to use UV irradiation to treat cancer directly, rather it is to use UV/Vis light irradiation to enhance the cytosolic transport of the azobenzene-based LNM delivery system. We demonstrated an application using this delivery system to enhance the antigen presentation in bone marrow derived dendritic cells (BMDCs), and then evaluated the effectiveness of using such primed BMDCs as adoptive cell therapy in a melanoma model.

Using autologous dendritic cells (DCs) loaded with engineered tumour antigens (Provenge®) was the first FDA approved cancer vaccine for treatment of prostate cancer. Hence using DC based adoptive cell therapy has clinical translation potential. Even though the melanoma model we demonstrated in this manuscript is a skin cancer, DC based cancer vaccines can be useful to treat other type of cancers since it is an immunotherapy, as shown in the case of Provenge® to treat prostate cancer.

We do appreciate the reviewer raising the question whether our approach can be effective in other tumour models. We conducted a pilot study using BMDCs primed with tumour lysates from 4T1 breast cancer cells (instead of the model antigen OVA shown in the melanoma model) using LNMs and evaluated its effectiveness in preventing tumour growth in a 4T1 breast cancer model in mice. In this study, tumour lysate proteins (TLPs) were isolated from 4T1-Luc (4T1 cells express luciferase) tumour cells, then mixed with TAzo-LNMs to obtain TAzo-LNM/TLP complexes. BMDCs were treated by TAzo-LNM/TLP complexes in the presence or absence of UV/Vis light irradiation (denoted as TAzo-LNM/TLP + UV/Vis and TAzo-LNM/TLP, respectively). We evaluated the TLP primed BMDCs in suppressing tumour growth in the subcutaneous (s.c.) breast cancer model. We first established the s.c. breast cancer model in BALB/c mice by s.c. injection of 1×10^6 4T1-luc cells. These 4T1-Luc tumour-bearing mice were then injected subcutaneously near the tumour sites with PBS or different BMDCs. Then the size of the tumours was measured using calipers. As shown in **Fig. 18a** and **18b**, BMDCs pre-treated with TAzo-LNM/TLP + UV/Vis effectively suppressed the growth of 4T1-Luc tumours and prolonged survival time of the mice compared with TAzo-LNM/TLP and PBS groups.

Since this manuscript mainly focuses on the concept of opening endo-lysosomes through

nanomechanical action, tumour therapy is used as an example to demonstrate its application. Hence, we decided not to include this piece of data to avoid distraction from our main objective. The investigation on whole tumor lysate primed DC, and a more detailed in vivo study will be published in future.

Supplementary Figure S18. a) Average tumour growth kinetics. b) Survival of 4T1-Luc tumour-bearing BALB/c mice. (Data for reviewing only)

4. In fig 4g, where the mice are irradiated after the SC injection, it will be worthwhile to investigate if other non-specific effects are seen on the other cellular compartments or other organs in the in vivo conditions.

Response: Thank you for your comments. During light irradiation, we tried to avoid systemic exposure and focused on the injection site. Due to the limited penetration of UV light, we expect the organs and cells in other regions will not be affected.

5. Please include the experiments and details on the stability and the kinetics (half-life) of the SAzo and TAZo lipidoids inside the cell?

Response: Thank you for your suggestions. We agree it's important to thoroughly characterize these novel azobenzene lipidoids in terms of stability, distribution, and metabolism. As stated earlier, this current manuscript mainly focuses on the concept of opening endo-lysosomes through nanomechanical action. We will report these results in future experiments.

Still, we conducted experiments to assess the half-life of SAzo and TAZo lipidoids inside cells. We measured the UV-Vis absorption spectrum of HeLa cells treated with and without TAZo-LNMs. As shown in **Fig. S19a**, the cells treated with TAZo-LNMs showed the absorption peak at 360 nm which is the signature absorption of the azobenzene. The peak intensity corresponds to the amount of the azobenzene. We calculated the amount of azobenzene inside the cells based on the change of the peak intensity. As shown in **Fig. S19b**, SAzo and TAZo lipidoids showed a decrease intracellular concentration over time, which may be attributed to cell proliferation and exocytosis. From this study, the half-life of SAzo and TAZo inside the cells are $\tau_{1/2} = 10.37 \pm 2.24$ h and $\tau_{1/2} = 8.93 \pm 1.95$ h respectively.

Supplementary Figure S19. a) UV–Vis absorption spectrum of HeLa cell suspension and the suspension of HeLa cells that pre-incubated with SAzo- or TAzo-LNMs. b) Cellular SAzo- and TAzo-lipidoid levels at different time points. (Data for reviewing only)

6. **Considering that the lipidoids, azobenzene products and the lipid-based drug delivery has been studied and explored by different groups, please discuss the novelty of the study which is questionable where it stands now.**

Response: Thank you for your suggestions. Even though we and many other groups have extensively publications on using lipidoids as a delivery system, using the light irradiation to generate the nanomechanical action to enhance the endo-lysosome disruption and delivery efficiency is a novel concept. The data shown in this manuscript demonstrated the feasibility of using the combination of chemical (charge interaction) and mechanical (light induced rotation) approaches to enhance the cytosolic delivery.

This LNM-based biomimetic approach demonstrates several advantages:

- 1) A generic strategy for multiple biologics because LNMs have similar structure and preparation procedure to clinically used lipid nanoparticles, which make this strategy applicable to a wide range of therapeutic nucleic acids and proteins;
- 2) Easy operation because irradiation is one of the most convenient energy inputs to drive LNMs.

7. **The translational significance and shortcomings of the approach should be also discussed.**

Response: Thank you for your suggestions. The discussion of both translational significance and limitations of LNM-based biomimetic approach were included in the Conclusion section (Page 15) in the revised manuscript.

REVIEWERS' COMMENTS

Reviewer #1 (Remarks to the Author):

The authors have well addressed all of the review's concerns.

Reviewer #2 (Remarks to the Author):

This revised manuscript has addressed our concerns.

Reviewer #3 (Remarks to the Author):

The authors were very responsive to my comments/suggestions adding additional experiments to further prove endosomal destabilization and the cytosolic effects on the innate immune response

Reviewer #4 (Remarks to the Author):

Authors have successfully addressed the concerns and revised the manuscript. I recommend it for publication.